# R2R: Efficiently Navigating Divergent Reasoning Paths with Small-Large Model Token Routing

**Tianyu Fu**[*1,2], **Yi Ge**[*1], **Yichen You**[1], **Enshu Liu**[1], **Zhihang Yuan**[2],
**Guohao Dai**[3,2], **Shengen Yan**[2], **Huazhong Yang**[1], **Yu Wang**[†1]

[1]Tsinghua University    [2]Infinigence AI    [3]Shanghai Jiao Tong University

## Abstract

Large Language Models (LLMs) achieve impressive reasoning capabilities at the cost of substantial inference overhead, posing substantial deployment challenges. Although distilled Small Language Models (SLMs) significantly enhance efficiency, their performance suffers as they fail to follow LLMs' reasoning paths. Luckily, we reveal that only a small fraction of tokens genuinely diverge reasoning paths between LLMs and SLMs. Most generated tokens are either identical or exhibit neutral differences, such as minor variations in abbreviations or expressions. Leveraging this insight, we introduce **Roads to Rome (R2R)**, a neural token routing method that selectively utilizes LLMs only for these critical, path-divergent tokens, while leaving the majority of token generation to the SLM. We also develop an automatic data generation pipeline that identifies divergent tokens and generates token-level routing labels to train the lightweight router. We apply R2R to combine R1-1.5B and R1-32B models from the DeepSeek family, and evaluate on challenging math, coding, and QA benchmarks. With an average activated parameter size of 5.6B, R2R surpasses the average accuracy of R1-7B by $1.6\times$, outperforming even the R1-14B model. Compared to R1-32B, it delivers a $2.8\times$ wall-clock speedup with comparable performance, advancing the Pareto frontier of test-time scaling efficiency. Our code is available at `https://github.com/thu-nics/R2R`.

## 1 Introduction

Large Language Models (LLMs) demonstrate strong capabilities across a wide range of tasks [1–3]. Building upon the largest and strongest LLMs, test-time scaling has become a prominent way to further boost their abilities on challenging tasks [4–7]. It is typically done by generating extensive Chain-of-Thought (CoT) reasoning before producing the final answer. However, this approach requires massive LLMs with hundreds of billions of parameters to generate thousands of tokens per query [8], resulting in significant inference overhead [9].

Distilled Small Language Models (SLMs), containing only a few billion parameters, offer much higher generation efficiency. Through supervised finetuning on LLM responses, SLMs can mimic LLM reasoning behaviors, making them a popular alternative. However, SLMs may still produce different reasoning paths from their LLM counterparts during inference, causing severe performance degradation. For example, compared to the R1-32B LLM, the R1-1.5B SLM provides different final answers on 45% questions in the AIME benchmark [10], suffering a $4.8\times$ reduction in accuracy as shown in Table 2.

---

[*]Equal contribution.
[†]Corresponding author: Yu Wang (yu-wang@tsinghua.edu.cn).

39th Conference on Neural Information Processing Systems (NeurIPS 2025).

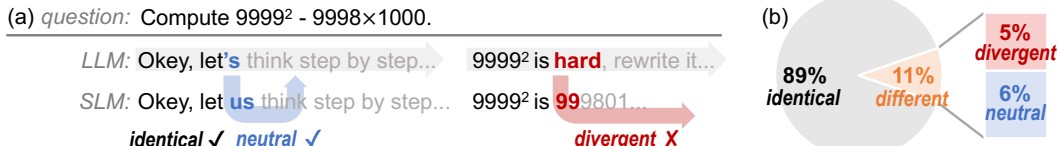

Figure 1: (a) Examples of R2R routing objective. Given a partial response as context, if SLM next-token prediction is not *identical* with LLM's, it is further categorized as *neutral* or *divergent* based on their effects on the reasoning path. (b) Distribution of *identical*, *neutral* and *divergent* labels in the R2R training set with 7.6M token labels.

Fortunately, we find that SLMs and LLMs often agree on next-token predictions given the same context. Instead, large performance gaps between them primarily arise from cumulative errors: their reasoning paths increasingly drift apart after some crucial differences in partial responses. To investigate this, we treat each step's LLM partial response as a prefix context and assess whether the next-token prediction from SLM is *identical* to LLM (Figure 1(a)). Across 2,094 queries totaling 7.6M tokens generated by the 32B LLM, the 1.5B SLM differs on only 11% of tokens—far less frequent than differences observed in final answers. Moreover, some of these differences are merely *neutral* variations, such as abbreviations or alternative expressions (e.g., *let's* vs. *let us*), which do not affect reasoning outcomes. The key drifts start from a subset of different tokens, which we call *divergent* tokens. These tokens genuinely alter the meaning, logic, or conclusion of the current sentence, thus diverging the subsequent reasoning path. This observation motivates us to selectively use SLM and LLM for different generation steps. It naturally leads to a critical research question:

*Can SLMs follow LLM reasoning paths by replacing only divergent tokens?*

If addressed, we could unlock substantial efficiency advantage of SLMs for most generation steps, yet preserving the high-quality reasoning typical of LLMs. This can enable better test-time scaling by advancing the efficiency-performance Pareto frontier.

The main challenge of SLM-LLM mix inference involves two interconnected parts: labeling the preferred model under certain objective, and designing the routing scheme to enforce it during inference. Previous methods typically route at the query level, selecting either SLM or LLM for entire response to maximize human preference win-rate within a cost budget [11, 12]. However, these approaches rely on human annotations and complex router designs, whose data labeling and routing scheme are both too expensive for fine-grained, token-level routing. Alternatively, speculative decoding methods aim for *identical* outputs between SLM and LLM at the token level [13–16]. They draft outputs with SLMs (or draft models) and periodically verify them with LLMs. While accurate, this strict verification leads to low acceptance rates. Additionally, mid-draft differences invalidate all subsequent tokens, severely restricting the accepted lengths as shown in Figure 2(b).

To address these challenges, we propose **Roads to Rome (R2R)**, a token-level routing method that selectively utilizes LLMs only for path-divergent tokens during SLM generation. We begin by automating token-level model preference labeling under a path-following objective. Starting from the LLM's reasoning paths, we identify *different* predictions for SLM and LLM, briefly continue generation from the point of difference, then use another LLM as verifier to determine whether the difference is truly *divergent* or merely a *neutral* variation. This labeling approach minimizes the lower bound of LLM usage by allowing neutral SLM-LLM differences. Using the resulting labeled dataset, we train a lightweight neural router to predict and immediately route divergent SLM tokens to the LLM for correction. We further improve routing accuracy by identifying predictive indicators of divergence such as SLM uncertainty and token rarity, available directly during SLM inference. Our contributions are summarized as follows.

- **Data Labeling Pipeline.** We develop an automatic pipeline to label divergent tokens. We formalize the global token-routing optimization problem, then propose a path-following strategy to generate routing labels with highly parallel, local decisions. We validate that SLM can effectively match LLM reasoning quality by following these routing labels.
- **Token-Router Design.** We introduce a token-level routing scheme using a lightweight neural router. We investigate SLM outputs that aid accurate token routing and incorporate them into the router, enabling immediate and more accurate routing of divergent tokens.

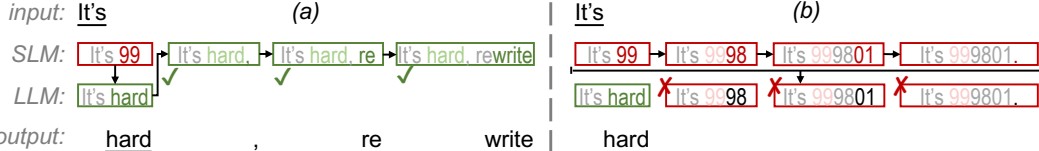

Figure 2: (a) R2R uses neural router to inspect SLM outputs at each step, **immediately corrects** divergent tokens with LLM, then continues generation from the corrected outputs. (b) Speculative decoding uses LLM to **periodically verify** if SLM outputs are identical to LLM predictions, invalidating all tokens after the first correction within the period.

- **Performance-Efficiency Pareto Frontier.** R2R enables more efficient performance scaling at test time. Compared to query-level routing and distilled R1-14B, it delivers 1.1–1.5× higher AIME accuracy with 1.5–1.6× lower latency. R2R also provides a 2.8× wall-clock speedup over R1-32B LLM at similar accuracy, while raising R1-1.5B SLM accuracy by 4.6× with only 12.9% LLM usage.

## 2   Related Work

Test-time scaling improves LLM performance at the higher cost of inference, often through the explicit generation of the CoT reasoning paths [4, 6]. For more effective scaling, previous works optimize the length and width of reasoning paths, or reduce the generation overhead of each path [17, 18].

**Controlling reasoning paths**.  Some approaches reduce LLM output lengths.  They employ prompting [19], post-training [20], or heuristics [21] to generate concise CoT with fewer decoding steps [19, 21]. Others explore the width of paths. They let LLMs generate multiple reasoning paths in parallel, then select the best outcome with methods like best-of-N voting [22] or external verifiers [23]. Both strategies modify the structure of reasoning paths, which are perpendicular to R2R's focus on reducing the overhead of each path.

**Model routing**. Model routing reduces generation cost by selecting the most suitable model for each query based on difficulty and budget.  Current works explore selection criteria of learned human preferences [11], reward signals [24], query tags [25], and model profiles [12]. Despite simplicity, they enforce the same LLM for each response, yielding suboptimal performance for the common mixed-difficulty generations. In contrast, R2R routes at token level to further improve efficiency.

**Speculative decoding**. Speculative decoding accelerates generation by fixing the low parallelism in LLM decoding [13–16]. It drafts outputs through SLM sequential decoding, then periodically verifies them with high-parallel LLM prefilling. However, speculative decoding pursues *identical* output token (distribution) between SLM and LLM, causing low acceptance rate. In addition, it is often that not all tokens generated by SLM within one draft-verify cycle can pass LLM verification. The mid-draft rejection invalidates all subsequent drafts and LLM verifications as shown in Figure 2(b), leading to frequent rollbacks. Expanding the single draft-chain to draft-tree alleviates the problem, but also incurs higher overheads that harm batch serving efficiency [16]. Considering the internality of CoT process, R2R accepts neutrally different output tokens, and immediately corrects all divergent tokens to avoid any rollback.

## 3   Model Preference Labeling

In Section 3.1, we formalize the token-level routing problem, aiming to minimize generation cost without sacrificing response quality. In Section 3.2, we introduce the path-following strategy for this problem, which assigns model preference labels to each output token, and empirically validate its effectiveness.

## 3.1 Token-level Routing Formulation

For autoregressive language models, reasoning can be represented as a sequence of next-token predictions. Throughout this paper, we focus on greedy sampling for simplicity:

$$y_i = \arg\max_y \mathcal{P}_{m_i}(y|x_0, \ldots, x_{n-1}, y_0, \ldots, y_{i-1}) = \arg\max_y \mathcal{P}_{m_i}(y|S_{<i}). \tag{1}$$

Here, $x_i$ and $y_i$ denote the input and output tokens, respectively. For notational simplicity, we define the token sequence at step $i$ as $S_{<i} = [x_0, \ldots, x_{n-1}, y_0, \ldots, y_{i-1}]$, where $S_{<0}$ is the input tokens. The next-token probability $\mathcal{P}_{m_i}$ is predicted by model $m_i \in \{\theta_s, \theta_l\}$ at step $i$, where $\theta_s$ and $\theta_l$ denote the SLM and LLM, respectively.

The essence of the routing strategy is to define a routing function $\mathcal{R}$ that selects the model for each decoding step:

$$m_i = \mathcal{R}(S_{<i}, \theta_s, \theta_l) \tag{2}$$

Our objective is to minimize the total generation *cost* while ensuring that the output sequence matches the *quality* of LLM-only outputs. We define the cost $\mathcal{C}$ as the sum of activated model parameters per token over the entire generation process. The *quality* of a response is evaluated by task-specific criteria, such as correctness for math problems, pass rates for coding tasks, or LLM-based grading for writing tasks. We define $\mathcal{V}$ as the verifier function, which returns 1 if and only if two sequences are of equivalent quality.

## 3.2 Path-following Routing Strategy

Optimally solving the token-level routing problem is computationally prohibitive, especially for large-scale data generation. While better routing sequences—potentially diverging from the LLM's reasoning path—may exist, finding them requires exhaustively searching a vast $O(2^n)$ space and generating thousands of output tokens for each search.

To overcome this practical limitation, we propose a greedy, sentence-level path-following routing strategy that reduces the search complexity to $O(n)$. Rather than exploring all possible model choices, our approach incrementally aligns mixed-model generation with the reasoning path established by the LLM. At each generation step, the strategy prefers the efficient SLM unless this would cause a meaningful divergence from the LLM's intended reasoning path, as determined by a continuation-and-verification mechanism.

Specifically, at each step, we first compare the next-token predictions from the SLM and LLM. If the predictions are *identical*, we confidently select the SLM, as this does not affect the output sequence. When predictions differ, we must determine whether the difference is *neutral* or *divergent*. To do so, we construct two candidate sequences by appending predictions from SLM and LLM to the previous token sequence, respectively. Both sequences are then continued using the LLM until a stopping criterion is met (e.g., EOS token is generated). These continuations reveal how the initial token difference affects subsequent reasoning, measured under optimal yet achievable conditions (i.e., LLM-only continuation). If the first continuation still matches the quality of the second under the verifier function $\mathcal{V}$, the difference is considered *neutral*; otherwise, it is *divergent* and the token is routed to the LLM.

$$m_i = \begin{cases} \theta_s, & \underbrace{y_i(\theta_s|S_{<i}) = y_i(\theta_l|S_{<i})}_{identical} \text{ or } \underbrace{\mathcal{V}(\mathcal{S}_s, \mathcal{S}_l) = 1}_{neutral} \\ \theta_l, & \underbrace{\mathcal{V}(\mathcal{S}_s, \mathcal{S}_l) = 0}_{divergent} \end{cases} \tag{3}$$

$$\mathcal{S}_s = S_{<i} \oplus \underbrace{[y_i(\theta_s|S_{<i})]}_{\text{SLM token}} \oplus \underbrace{[y_{i+1}(\theta_l|S_{<i} \oplus [y_i(\theta_s|S_{<i})]), \ldots, \text{EOS}]}_{\text{LLM continuation}} \tag{4}$$

$$\mathcal{S}_l = S_{<i} \oplus \underbrace{[y_i(\theta_l|S_{<i})]}_{\text{LLM token}} \oplus \underbrace{[y_{i+1}(\theta_l|S_{<i} \oplus [y_i(\theta_l|S_{<i})]), \ldots, \text{EOS}]}_{\text{LLM continuation}} \tag{5}$$

Equations 3–5 formalize the routing strategy. Here, $y_i(m_i|S_{<i})$ indicates that this output token is generated by model $m_i$ given the previous sequence $S_{<i}$, as a simplified expression of Equation 1. The continuation sequences, respectively generated after SLM and LLM token, are denoted by $\mathcal{S}_s$ and $\mathcal{S}_l$. The operator $\oplus$ indicates concatenation of token sequences.

Table 1: Statistics of tokens difference and divergence across query types in the training dataset.

| Type | #Query | #Token | #Different | Diff. Rate | #Divergent | Div. Rate |
|---|---|---|---|---|---|---|
| Math | 587 | 2.9M | 195.1K | 6.8% | 81.8K | 2.8% |
| Code | 698 | 3.2M | 329.0K | 10.3% | 151.9K | 4.7% |
| QA | 735 | 1.4M | 290.8K | 20.2% | 139.4K | 9.7% |
| Summary | 2094 | 7.6M | 814.9K | 10.8% | 373.1K | 4.9% |

When continuation produces the full response by stopping only at the regular EOS token, we call this **full path-following** routing. By using the quality verifier from Section 3.1, the mixed-generated token sequence is guaranteed to achieve the same *quality* as its LLM-only counterpart, as it always remains on a path that could achieve LLM-only quality. The formal proof of this quality guarantee is provided in Appendix E. While the resulting model choice sequence $M_{<i} = [m_0 \ldots m_{i-1}]$ can be used as labels for router training, full continuation is computationally expensive for large-scale data generation. In addition, the effect of current difference to the final output quality thousands of tokens away is too hard to learn for the neural router to be trained.

In practice, we use **sentence-level path-following** routing, where the continuation ends at the current sentence, as shown in Figure 3 (step 2). We monitor sentence-ending symbols, like period, during continuation and use existing semantical sentence separators [26, 27] to conclude generation if the sentence truly ends. To verify this local continuation, a capable LLM serves as a sentence-level verifier $\mathcal{V}'$, as shown in Figure 3 (step 3). It is prompted to compare the continuations and determine whether the initial token difference introduces a meaningful *divergence* from the LLM's intended reasoning path, or merely a *neutral* abbreviation. Instead of verifying the entire generation, this approach checks the reasoning path at the sentence level, greatly improving data labeling efficiency.

We empirically validate the effectiveness of sentence-level path-following routing using Qwen2.5-72B [28] as the verifier model, with prompts detailed in Appendix F.1. Among 17 AIME-24 questions correctly solved by R1-32B within an 8K-token limit, our path-following strategy achieves comparable accuracy (16 questions correctly answered) while relying on the larger R1-32B model for only 3% of generated tokens.

By locally evaluating token choices through sentence-level path-following routing, we closely align mixed inference with the LLM's high-quality reasoning path, eliminating the prohibitive overhead of global evaluations. However, direct use of this strategy for real-time inference is impractical, as it relies on costly LLM continuation and verification. Instead, the local nature of our strategy simplifies routing decisions, creating an easier learning task for a neural router compared to global routing. We therefore design and train a lightweight neural router that efficiently approximates this strategy, relying solely on SLM outputs to determine when to use the LLM during inference.

## 4 Token-Level Neural Router

This section describes our methodology for constructing the neural router. Specifically, we detail how routing labels are generated for training using the sentence-level path-following strategy (Section 4.1), identify predictive SLM indicators for the router (Section 4.2), and outline the neural router's architecture along with its routing scheme for inference deployment (Section 4.3).

### 4.1 Training Data Generation

We use sentence-level path-following routing to generate training labels for the neural router, incorporating several optimizations to control data labeling overhead.

Figure 3 shows our data generation pipeline. Given queries from existing datasets, we first obtain the complete LLM response, either directly from the dataset or via batched LLM inference. Next, we use highly parallel SLM prefilling to efficiently identify tokens where the SLM prediction is *identical* to the LLM, allowing us to exclude about 90% of tokens from further processing. For the remaining 10% of differing tokens, we perform batched LLM continuations from each SLM prediction. To further improve efficiency, we apply prefix caching in current frameworks [29, 30] to reuse KV-Cache

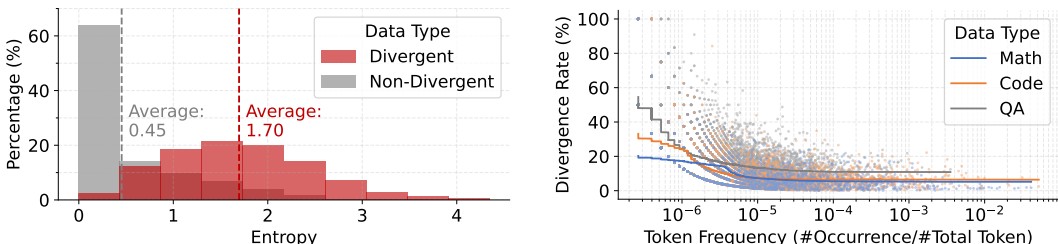

| query | Compute $9999^2 - 9998 \times 1000$. | | | | | | | | | | | | | | |
|---|---|---|---|---|---|---|---|---|---|---|---|---|---|---|---|
| **step0:** LLM response | Let | 's | think | step | by | step | . | 99 | 99 | $^2$ | is | hard | , | re | write | it |
| **step1:** SLM prefill | Let | **① us** different | think | step | by | step | . | 99 | 99 | $^2$ | is | **② 99** different | , | re | write | it |

| **step2:** LLM continuation | ① Let | **us** $\longrightarrow$ think about it step by step. |
| | ② Let | 's think step by step . 99 99 $^2$ is **99** $\longrightarrow$ 9801. |

| **step3:** verify | ① Verify | Let's think step by step | **and** | Let **us** think about it step by step | ☐ neutral |
| | ② Verify | $9999^2$ is hard, rewrite it | **and** | $9999^2$ is **99**9801 | ☐ divergent |

| output label | SLM | SLM | SLM | SLM | SLM | SLM | SLM | SLM | SLM | SLM | SLM | LLM | SLM | SLM | SLM | SLM |

Figure 3: R2R data labeling pipeline. Given a query question, the LLM first generates a response to establish the desired reasoning path. The SLM then prefills this path to identify *identical* and *different* next-token predictions. For each *different* SLM token, the LLM continues generation from that point. Finally, a verifier model determines whether each difference leads to a *neutral* or *divergent* outcome, labeling the model preference as SLM or LLM, respectively.

Figure 4: Oracle insights for router design. (a) SLM entropy distribution, clipped at 99th percentile for visualization clarity (b) Divergence rate and frequency of different tokens.

computations for shared context prefixes across multiple continuations (e.g., everything preceding *Let* in Figure 3). Continuations for the corresponding LLM tokens, $\mathcal{S}(\theta_l)$, are directly extracted from the pre-generated LLM response, eliminating redundant computation. Finally, the verifier model compares both continuations and label routing preference. Further analysis of the robustness of the verifier and the comparison to human experts are provided in Appendix B.1.

Using this pipeline, we efficiently generate 7.6 million routing labels in approximately 2.3 days on 8 A800 GPUs, covering topics of math, coding, and QA with queries from the Bespoke-Stratos [31] dataset. Table 1 summarizes the statistics of the generated training dataset.

## 4.2 Predictive Indicators of Divergence

We explore predictive indicators that can help identify divergent tokens. To enable immediate routing, we focus on indicators that can be acquired solely during the SLM's next-token predictions. The following analysis is based on 7.6 million tokens in our training set.

**SLM logits**. As shown in Figure 4(a), divergent tokens exhibit substantially higher entropy in the SLM's output logits, with a $3.8\times$ mean value over that of non-divergent tokens. We observe similar trends with other uncertainty measures [32], and concurrent work [33] also confirms such observation. These empirical results indicate that increased uncertainty in SLM predictions is strongly correlated with token divergence. Motivated by this, our router takes top-100 SLM logit values as one of its input features.

**Token frequency**. Figure 4(b) shows that low-frequency tokens in the dataset are more likely to be divergent. This likely arises from the long-tail token distribution in the training data, making rare tokens harder for SLMs to model effectively due to the limited capacity [34]. Given this insight, our router explicitly incorporates token-frequency biases by using the token embedding of as router inputs.

### 4.3 Router Design and Routing Scheme

**Model architecture**. Guided by insights in Section 4.2, we design the neural router as a lightweight, six-layer feed-forward network (FFN) with 56M parameters. It takes the SLM's output logits and tokenized embedding, along with its last-layer hidden states for additional semantic context. All inputs are linearly projected, concatenated, and fed into the FFN backbone. The router outputs a binary classification probability, indicating whether the current token diverges from the LLM's reasoning path. Full network architecture detail descriptions are in Appendix A.1.

**Training scheme**. We train the router with cross-entropy loss using the labeled data described in Section 4.1. To address class imbalance caused by the low divergence rate, we re-weight the loss inversely to class frequency. After training, we use the validation set to select the routing threshold that meets the user-defined LLM usage rate. The full training details are provided in Appendices A.3 and A.2.

**Routing scheme**. Unlike speculative decoding methods that periodically verify SLM outputs, our routing scheme aims to immediately decide whether to accept each SLM token, eliminating the need for rollbacks. As shown in Figure 2, this approach reduces unnecessary draft and verification computations, which is especially beneficial in computation-intensive batch-serving scenarios. Specifically, the neural router estimates divergence probability at each generation step using SLM outputs. When this probability exceeds a predefined threshold $p_{th}$, the LLM is invoked to correct the current output token. Following speculative decoding methods [13, 15], we utilize highly parallel prefilling for efficient LLM KV-Cache updates, whose overhead can be further reduced by overlapping them with SLM decoding [35].

## 5 Experiment

### 5.1 Setup

**Baselines**. We use DeepSeek-R1-Distill-Qwen models as baselines, denoted by R1-$M$B, where $M$ indicates the model size in billions. We designate R1-1.5B and R1-32B as SLM and LLM, respectively, while intermediate sizes (7B, 14B) capture distillation scaling behavior. We compare various query-level routing (QR) methods from the RouteLLM framework [11], including similarity-weighted ranking (QR-SW), matrix factorization (QR-MF), a BERT-based classifier (QR-BERT), and a Llama3-8B-based classifier (QR-LLM). For speculative decoding, we adopt EAGLE2 [14] and HASS [15] with R1-32B LLM. We use the official HASS draft model, and train the EAGLE2 draft model using its official script, as no pre-trained EAGLE2 draft is provided for R1-32B.

**R2R setup**. R2R routes between R1-1.5B and R1-32B using a lightweight 56M-parameter FFN router, trained on 7.6M token-level routing labels described in Section 4.1. More details on router architecture, training data, hyperparameters, and implementation are presented in Appendix A. Note that the router weights are fixed for all evaluations. The routing threshold $p_{th}$ is selected for 6B average parameter usage on the validation set. Performance-efficiency trade-offs are controlled solely by adjusting $p_{th}$, without retraining the router.

**Benchmark**. We evaluate methods across challenging reasoning benchmarks, including mathematics (AIME 2024–2025 [10]; denoted as AIME), graduate-level question-answering (GPQA-Diamond [36]; denoted as GPQA), and coding tasks (LiveCodeBench 2024-08–2025-01; denoted as LiveCodeBench [37]). All experiments use a maximum output length of 32K tokens and zero generation temperature to ensure reproducibility.

**Efficiency metric**. Following previous works [2, 38], we use the *average activated parameters per token* as a hardware-agnostic efficiency metric, referred to as the *average parameter* ($\bar{M}$) for brevity. For query-level routing, $\bar{M}$ is computed as the weighted average of SLM and LLM parameters based on their activation ratios across all outputs. For R2R, $\bar{M}$ includes the SLM and router parameters, along with the LLM parameters weighted by the LLM activation ratio. We also report the total *Cost* ($C$), defined as average activated parameters multiplied by the average output tokens per query. The average parameter and total cost reflect the average decoding speed and total latency, respectively. In addition, we report hardware-specific decoding speed on NVIDIA A800-80GB GPUs using SGLang [29] framework.

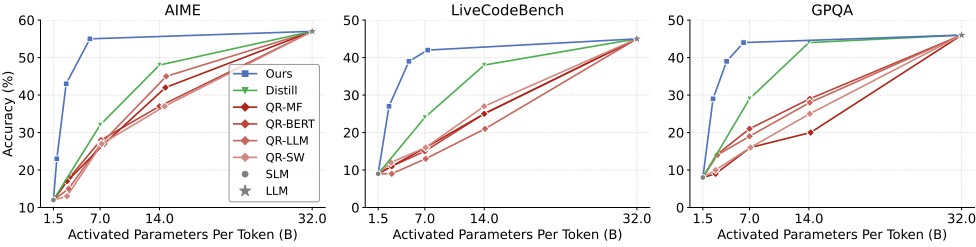

Figure 5: Scaling of accuracy versus average activated parameters per token, evaluated across AIME, GPQA, and LiveCodeBench. R2R advances the Pareto frontier beyond distillation and query-level routing methods.

Table 2: Performance and efficiency comparison across benchmarks and methods. *Param.* denotes the average activated parameters per token in billions; *Cost* is the average output tokens (thousands) per query multiplied by average parameters (billions) .

| Type | Method | AIME | | | LiveCodeBench | | | GPQA | | | Average | | |
|---|---|---|---|---|---|---|---|---|---|---|---|---|---|
| | | Acc. | Param. | Cost | Acc. | Param. | Cost | Acc. | Param. | Cost | Acc. | Param. | Cost |
| SLM | R1-1.5B | 12% | 1.5 | 42 | 9% | 1.5 | 43 | 8% | 1.5 | 42 | 10% | 1.5 | 42 |
| LLM | R1-32B | 57% | 32.0 | 487 | 45% | 32.0 | 606 | 46% | 32.0 | 519 | 50% | 32 | 537 |
| 7B | R1-7B | 32% | 7.0 | 148 | 24% | 7.0 | 168 | 29% | 7.0 | 147 | 28% | 7.0 | 154 |
| | QR-SW | 27% | 7.2 | 168 | 16% | 7.1 | 188 | 16% | 7.1 | 179 | 20% | 7.1 | 178 |
| | QR-LLM | 27% | 7.1 | 170 | 13% | 7.1 | 195 | 19% | 7.0 | 172 | 20% | 7.1 | 179 |
| | QR-BERT | 28% | 7.1 | 160 | 15% | 7.0 | 189 | 21% | 7.0 | 169 | 21% | 7.0 | 173 |
| | QR-MF | 27% | 7.5 | 168 | 16% | 7.1 | 190 | 16% | 7.1 | 181 | 20% | 7.2 | 180 |
| 14B | R1-14B | 48% | 14.0 | 239 | 38% | 14.0 | 267 | 44% | 14.0 | 197 | 43% | 14.0 | 234 |
| | QR-SW | 37% | 14.5 | 295 | 27% | 14.0 | 333 | 25% | 14.1 | 318 | 30% | 14.2 | 315 |
| | QR-LLM | 45% | 14.8 | 277 | 21% | 14.1 | 356 | 28% | 14.1 | 299 | 31% | 14.3 | 311 |
| | QR-BERT | 37% | 14.0 | 280 | 25% | 14.0 | 342 | 29% | 14.1 | 297 | 30% | 14.0 | 306 |
| | QR-MF | 42% | 14.7 | 284 | 25% | 14.0 | 336 | 20% | 14.2 | 338 | 29% | 14.3 | 319 |
| R2R | Ours | **55%** | **5.5** | **101** | **39%** | **5.1** | **106** | **44%** | **6.3** | **101** | **46%** | **5.6** | **103** |

## 5.2 Performance

**Scaling behavior**. Figure 5 shows accuracy scaling with average activated parameters per token. Query-level routing (QR) methods exhibit near-linear accuracy scaling from 1.5B to 32B parameters. Distilled models (R1-7B, R1-14B) achieve superlinear gains with extensive training, reaching 88% of R1-32B's accuracy with just 50% of the parameter size at 14B. By routing only divergent tokens to the LLM, R2R achieves 92% average accuracy with only 17% average parameters, delivers even better scaling at a new Pareto frontier. Moreover, due to reduced output lengths, R2R offers an even better trade-off in terms of accuracy versus total test-time cost $C$ (see Appendix B.7). The routing threshold in R2R also enables flexible, post-training control of this trade-off.

**Numerical comparison**. Table 2 shows numerical details of model performance around average parameter sizes of 7B and 14B. With an average parameter size of 5.6B, R2R outperforms the best query-level routing methods (in both 7B and 14B) by 1.4–2.4× and 1.2–1.4×, respectively. Compared to distilled models, R2R improves accuracy by 1.4–1.7× over R1-7B and even surpasses R1-14B in average accuracy by 1.1×. Relative to the extremes, R2R achieves 4.6× higher accuracy than R1-1.5B and retains 92% of R1-32B's accuracy, while using the LLM for only 11–15% of tokens.

**Generalizability**. Beyond the R1 family, we further train and evaluate R2R on the Qwen3 series (both dense and MoE variants) spanning 0.6B, 1.7B, 8B, 30B-A3B, and 32B parameter scales. Across all configurations, R2R consistently surpasses both the base Qwen3 models and query-level routing baselines, demonstrating strong adaptability and cross-architecture generalization. We also evaluate on Arena-Hard (dialogue) and MMLU-Redux-Philosophy benchmarks, which are beyond the selected mathematical reasoning, QA, and code generation tasks. Remarkably, R2R continues to outperform

| Method | #Token(K) | Latency(s) | Speed(tok/s) |
|---|---|---|---|
| R1-1.5B | 28.2 ±10.5 | 199 ±81 | 141.6 |
| R1-14B | 17.1 ±12.3 | 328 ±272 | 52.1 |
| R1-32B | 15.2 ±12.4 | 498 ±456 | 30.5 |
| QR-SW | 20.3 ±13.5 | 336 ±379 | 55.5 |
| QR-LLM | 18.7 ±13.5 | 332 ±334 | 56.1 |
| QR-BERT | 19.9 ±13.2 | 350 ±367 | 57.0 |
| QR-MF | 19.3 ±13.3 | 347 ±359 | 55.6 |
| Eagle2 | 17.4 ±13.1 | 244 ±194 | 71.4 |
| HASS | 18.8 ±12.9 | 256 ±197 | 73.3 |
| **Ours** | 18.4 ±13.5 | **218** ±161 | **84.3** |

Table 3: Comparison of latency, output token length, and average speed across methods. Subscripts note the standard deviations across AIME.

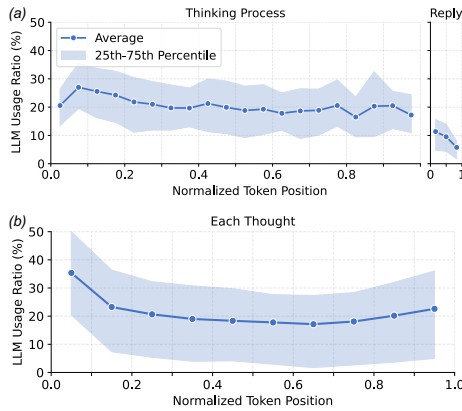

Figure 6: LLM usage rate at different positions, normalized by (a) thinking and reply process, (b) each thought.

R1-14B while maintaining an average activated parameter size of only 6.1–6.7B, confirming its robust generalization across domains and model families (see Appendix B.2.2).

**Sampling method extension**. Beyond greedy decoding, we extend R2R to nucleus (top-p = 0.95) with temperature (temperature = 0.6). R2R still greatly advances Pareto frontier, with 1.4-1.5x higher AIME score over query-level routing, reaching R1-14B score with only 8.6B average parameters. Implementation details and theoretical analysis are provided in Appendix C.1.

### 5.3 Efficiency

**Wall-clock speedup**. Table 3 reports the wall-clock latency and speed for all methods on the AIME benchmark. All baselines use the official, highly efficient SGLang [29] framework and are evaluated with tensor parallelism on two NVIDIA A800-80GB GPUs. R2R uses the same thresholds as in Table 2; query-level routing methods use the 14B version for comparable performance. R2R achieves 1.62× and 2.76× generation speed over R1-14B and R1-32B, respectively. Compared to query-level routing, R2R delivers 1.48–1.52× speedup. It also outperforms highly optimized speculative decoding methods with tree-like drafts, which speedup mostly at the current single-batch setup [16]. Further system-level optimization can be done to yield even greater gains for R2R. Note that, in theory, speculative decoding should exactly match the R1-32B LLM's output. However, we occasionally observe inconsistencies for very long outputs, likely due to cumulative numerical errors. We faithfully report this observation without questioning the equivalence guarantee of speculative decoding.

**Computation and memory access**. Compared with the LLM baseline, R2R achieves a 5.4× reduction in per-token memory access. The total computation increases only marginally due to the additional SLM and router computation. Because decoding is largely memory-bound, the reduced memory traffic yields higher throughput. This substantial reduction in memory traffic translates into higher throughput that outweighs the minor compute overhead. Compared with speculative decoding approaches such as Eagle2 and HASS, R2R requires about 17.0× less total computation, primarily because its immediate correction prevents the LLM from repeatedly prefilling unused tokens during periodic verification (see Figure 2). At the same time, it reduces memory access by 2.4–2.5× relative to these speculative methods, yielding a more balanced compute–memory trade-off. More details are provided in Appendix B.3.2.

### 5.4 Ablation Study

Starting from R2R in the first row of Table 4, we evaluate the effectiveness of our design by retraining the router with alternative objectives or reduced inputs. We adjust the routing thresholds $p_{th}$ of ablation variants to match or slightly exceed the LLM rate of our original router. All experiments are conducted on the AIME benchmark with all other settings held constant. Further discussions on the trade-off between the LLM usage rate and recall are provided in Appendix B.5.3.

Table 4: Ablation study on routing objectives and router inputs. *HS* denotes last-layer hidden states; *Token* denotes token embedding. Recall denotes the recall rate of divergent token on the validation dataset. Italicized words indicate ablation focuses.

| Objective | Router Input | Acc. | Recall | LLM Rate | #Token(K) | Param.(B) | Cost(KB) | Latency(s) |
|---|---|---|---|---|---|---|---|---|
| Divergent | HS+Token+Logits | 55% | 95% | 12.4% | 18.4 | 5.5 | 101 | 218 |
| *Different* | HS+Token+Logits | 40% | 88% | 13.1% | 21.0 | 5.7 | 119 | 228 |
| Divergent | *HS+Token* | 47% | 85% | 13.3% | 18.8 | 5.8 | 109 | 253 |
| | *HS* | 42% | 83% | 14.1% | 18.4 | 6.0 | 110 | 245 |

**Routing objective**. As discussed in Section 3, we categorize *different* next-token predictions as either *neutral* or *divergent*. R2R improves efficiency by tolerating *neutral* differences and only routing truly *divergent* tokens to the LLM. When the router is trained to use the LLM for all *different* tokens, it fails to reach the original accuracy within the same amount of LLM usage, facing $1.4\times$ accuracy degradation, as shown in the second row of Table 4. This confirms that restricting LLM usage to only divergent tokens is crucial for reducing cost while maintaining high accuracy.

**Router input**. As discussed in Section 4, both SLM logits and token embeddings are strong indicators of divergence to be used as router inputs. When gradually remove these features, accuracy drops by up to $1.3\times$, underscoring their importance. Note that while SLM logits can be computed from last-layer hidden states within the router in principle. However, doing so requires the capacity of the 234M-parameter embedding layer, which exceeds the capacity of the 56M-parameter neural router.

**SLM–LLM combination**. We extend R2R to the Qwen3 family by fixing the LLM and varying the SLM to form three combinations: Qwen3-0.6B+8B, Qwen3-1.7B+8B, and Qwen3-4B+8B. As Figure 10 in Appendix B.5.3 shows, with the LLM fixed, smaller SLM delivers a better Pareto frontier. It indicates that, when selecting SLM–LLM combinations, favoring a smaller SLM is often more efficient for the same accuracy target due to lower cost. We further analyze R2R alongside orthogonal techniques such as MoE and query-level routing in Appendix B.4.

## 5.5 Routing Result Observation

We analyze the routing behavior of R2R on the AIME benchmark, considering finished responses within the 32K token limit. Figure 6(a) shows the LLM usage rate across response positions. Each response is divided into the thinking process and the subsequent reply, with positions normalized to [0, 1]. The subplot widths reflect their respective average lengths. The results show that R2R routes noticeably fewer tokens to the LLM during the reply phase. It reflects the intuition that after internal thinking, the reply itself is straightforward and less demanding.

Following prior work [21], we further segment the thinking process into sequential thoughts based on tokens such as *Wait* and *Alternatively*. Figure 6(b) examines the LLM usage ratio within each thought. It shows that R2R relies more on the LLM at the beginning and end of each thought. This aligns with the expectation that the initial tokens set the direction for the thought, while the concluding tokens determine whether to end the thought, branch into alternatives, or continue deeper reasoning. Notably, these routing patterns are not hand-crafted but naturally emerge from router training. It helps R2R to effectively allocate LLMs for more better test-time scaling.

## 6 Conclusion

We introduce R2R, a token-level router that lets an SLM track an LLM's reasoning by correcting only path-divergent tokens. We propose a path-following labeling strategy and identify predictive signals that train a neural router for accurate token selection. On challenging benchmarks, R2R outperforms R1-14B with less than 7B average parameters, boosts SLM performance by $4.6\times$ with under 15% LLM usage, and achieves a $2.8\times$ wall-clock speedup over the LLM at comparable accuracy.

**Limitations**. Our routing strategy is chiefly tuned and tested for greedy sampling. While we have validated R2R on a limited set of alternative sampling policies, broader exploration could improve versatility. Further system-level optimizations are also needed to realize its full cost benefits.

## Acknowledgments and Disclosure of Funding

This work was supported by National Natural Science Foundation of China (No. 62325405, 62104128, U19B2019, U21B2031, 61832007, 62204164, 92364201), Tsinghua EE Xilinx AI Research Fund, and Beijing National Research Center for Information Science and Technology (BNRist). We thank Zinan Lin, Xuefei Ning, and Donglin Yang for their valuable discussions and suggestions. We thank Chao Xiong for his support with the SGLang interface. We also thank all the support from Infinigence-AI.

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

# A  Additional Experiment Setups

## A.1  Router Architecture

**Inputs projection**. At each decoding step, we use the hidden states of the last layer, the top 100 logits with the highest values and the embeddings of the predicted token from the SLM to generate the routing result. We first apply linear projections to align their dimensions with the hidden states and then concatenate the features from the logits, hidden states, and token embeddings. Finally, we use another linear layer to project the concatenated features to match the input feature dimension of the model backbone.

**Neural network backbone**. For the router architecture, we adopt a six-layer feed-forward network (FFN) with residual connections between blocks as the backbone, using a hidden size of 1024. The architecture of each FFN follows the common design used in previous LLMs [39]. Each block begins with LayerNorm for input normalization, followed by a pair of linear projections forming an expand-then-contract structure with an expansion factor of 4. Dropout is applied to each linear layer, with a GELU activation function between them. These blocks are connected using residual connections. At the end of the last block, we apply an additional layer normalization and a linear layer to convert the output to a single value, followed by a sigmoid function to produce the normalized prediction of the router. A predefined threshold $p_{\text{th}}$ between 0 and 1 is used for generating binary results from the router output. Predictions above the $p_{\text{th}}$ are considered that current tokens diverge from the LLM's reasoning path.

## A.2  Routing Data

**Training dataset**. Our training data for the router are sourced from tasks across three distinct scenarios: mathematics, code, and question answering (QA). The mathematics problems are drawn from the American Invitational Mathematics Examination (AIME) [10], covering the years 1983 to 2022. Code and QA problems are sampled from Bespoke-Stratos-17k dataset [31]. We use only the formatted questions from these datasets as prompts and generate responses using DeepSeek-R1-Distill-Qwen-32B, with the temperature set to 0 and a maximum generation length of 32,768 tokens. Only responses that contain an end-of-sequence (EOS) token within the specified length are retained as effective samples, which will be used for subsequent stages of our data generation pipeline, as discussed in Section 4.1.

**Validation dataset**. Our validation dataset are constructed in the exact same way as the training data, but with different queries. The validation dataset comprises all 30 problems from AIME 2023, 69 coding problems from the Bespoke-Stratos-17k dataset that are excluded from the training set, and 60 QA problems selected from the GPQA-Extended [36] dataset.

## A.3  Training Scheme

**Loss function**. Due to the significant class imbalance in the training data, we adopt the weighed BCEWithLogitsLoss as our loss function. The weight of each class is calculated inversely proportional to its frequency in the class, which encourages the model to pay more attention to underrepresented classes.

**Training hyperparameters**. During training, we employ the AdamW optimizer with hyperparameters $\beta_1 = 0.9$ and $\beta_2 = 0.999$. The learning rate is set to $5 \times 10^{-5}$, with a dropout rate of 0.1 and a weight decay of $5 \times 10^{-4}$. We train the neural network with float32 precision. The router is trained for up to 50 epochs using a batch size of 1024, with early stopping applied based on a patience of 10 epochs. Validation is performed at every epoch. We adopt the checkpoint corresponding to the best-performing epoch on the validation set as the final router used.

**Threshold selection**. After training, we use the validation dataset to select a preferred threshold. We pass the pre-collected neural router inputs from the validation dataset through the neural router and record the predicted divergence probabilities. By sweeping $p_{\text{th}}$ from 0 to 1, we analyze how different thresholds affect the LLM usage rate and average parameter size., as shown in Figure 8. This process is efficient, as all router inputs (SLM logits, token embeddings, and last-layer hidden states) are pre-collected and evaluated in a single pass. During inference, given any user-defined average parameter budget, we set the threshold to meet the target budget accordingly.

## A.4 Routing System Implementation

**Model initialization**. The routing system consists of three components: a SLM (R1-1.5B), a LLM (R1-32B), and a router model. The SLM is loaded onto a single GPU (GPU 1) using the SGLang scheduler, with the *mem_fraction_static* set to 0.15. The LLM employs tenser-parallel inference distributed across two GPUs (GPU 0 and GPU 1) via SGLang schedulers managed by PyTorch's distributed multiprocessing framework with the NCCL backend, with the *mem_fraction_static* set to 0.80. The router model is directly loaded onto GPU 0 using standard PyTorch, independent of the SGLang interface. Prior to inference, each of SLM and LLM is individually warmed up using simple inputs to stabilize GPU kernels and caches, ensuring consistent inference latency.

**Inference workflow**. During each inference step, the workflow begins with the SLM decoding a single token, returning the corresponding last-layer hidden states and output logits. The router generates a divergence probability based on these outputs of SLM. If the probability surpasses the predefined threshold, the LLM is activated to extend the sequence by one token. Specifically, a new request, constructed from the input token IDs, is placed into the LLM's input queue. Subsequently, a new schedule batch is initialized for the LLM, explicitly setting the forward mode to *EXTEND* and allocating appropriate memory for handling input sequences. The system maintains prefix indices to track processed tokens, enabling efficient token management between models. When the LLM extends a token, it is communicated back through an output queue to replace the SLM's predicted token. A token manager actively tracks the sequence states during the generation process, managing active sequences and handling termination conditions effectively. At each token position, the dynamic routing mechanism assesses model outputs, determines the appropriate routing decision, and updates sequence states accordingly. This iterative process continues until a sequence is completed or reaches the predefined maximum token limit.

# B Additional Experiment Results

## B.1 Verifier Robustness

### B.1.1 LLM Verifier Against Human Verifier

**Setup**. We conducted a human evaluation to validate the verifier's reliability. Specifically, we recruited four undergraduate students to independently label 1,357 *differences* as either *neutral* or *divergent*. The *differences* are between R1-1.5B and R1-32B on the first six AIME-2024 questions. Three annotators' labels determined the ground truth for *divergence* by majority voting (general *divergence*, positive rate: 24.8%) and unanimous consent (core *divergence*, positive rate: 11.9%). The fourth annotator's labels, along with several LLM verifiers, were then compared against these ground truths. All annotators and LLM verifiers received identical contexts and instructions.

**Results and discussion**. Our selected verifier (Qwen2.5-72B) closely matches human expert performance. The detailed results are shown in Table 5.

Table 5: Verifier performance against **majority vote** (General Divergence, Positive Rate: 24.8%) and **unanimous consent** (Core Divergence, Positive Rate: 11.9%) ground truths.

| Verifier | Majority Voting (General) | | | Unanimous Consent (Core) | | | Positive Rate |
|---|---|---|---|---|---|---|---|
| | Accuracy | Recall | Precision | Accuracy | Recall | Precision | |
| Human | 0.82 | 0.84 | 0.59 | 0.76 | 0.96 | 0.32 | 0.35 |
| Qwen2.5-72B | 0.84 | 0.88 | 0.62 | 0.76 | 0.97 | 0.33 | 0.35 |
| Qwen2.5-7B | 0.78 | 0.85 | 0.54 | 0.71 | 0.94 | 0.28 | 0.39 |
| Qwen2.5-3B | 0.75 | 0.69 | 0.50 | 0.72 | 0.75 | 0.26 | 0.34 |

### B.1.2 Impact of Verifier Quality

**Setup**. Next, we examined the sensitivity of our method to verifier quality. We applied the sentence-level path-following routing method (Equations 3–5) to the AIME 2024 benchmark across verifiers (Qwen2.5-3B to Qwen2.5-72B) within 8K token budget.

**Results and Discussion**. Our analysis reveals that even moderate-recall verifiers (e.g., Qwen2.5-3B) still significantly improve reasoning path guidance compared to the distilled 7B baseline. Note that the current verifier prompt is tuned for the Qwen2.5-72B verifier. Empirically, smaller verifiers should benefit from prompts biased towards classifying ambiguous cases as *divergent*, thereby enhancing recall.

Table 6: Impact of verifier quality on path-following routing performance on AIME 2024.

| Type | Model Param. (B) | Labeling Verifier | AIME'24 #Acc.@8K |
|---|---|---|---|
| LLM Baseline | 32 | - | 17 |
| SLM Baseline | 1.5 | - | 2 |
| Distilled | 7 | - | 8 |
| Path-following | 2.3 | Qwen2.5-72B | **16** |
| Path-following | 2.5 | Qwen2.5-7B | 12 |
| Path-following | 2.2 | Qwen2.5-3B | 11 |

## B.2 Generalizability

### B.2.1 Data Labeling Method Generalizability

The labeling method of R2R, relying on semantic comparison rather than formal verification, can easily generalize across tasks and achieve high performance even in some out-domain tasks. Unlike methods requiring task-specific external tools (e.g., formal proofs [40], code execution [41]), R2R uses an LLM verifier to identify general semantic divergence in meaning, reasoning, logic, or conclusions F.1.

In practice, we apply the identical labeling strategy and verifier prompt across closed-form math, coding tasks, and open-ended QA tasks. This consistency enables our router to naturally generalize to unseen tasks during inference, as demonstrated later.

For tasks with subjective divergence criteria that are challenging to identify semantically within a single sentence, the continuation length can be extended. This adjustment gradually transitions from sentence-level routing to full routing, the latter of which only requires overall response quality evaluation, which is a feasible requirement commonly met for LLM tasks.

### B.2.2 Router Generalizability

**Generalize across benchmarks**. Our router leverages outputs from the SLM (e.g., logits) to predict token divergence. Benefiting from the inherent generalizability of SLMs, indicators such as logits entropy robustly identify divergent tokens across different tasks. To validate the router's generalizability, we directly apply our router, trained for math, QA, and code, to additional benchmarks: Arena-Hard for Dialog [42], and the Philosophy split from MMLU-Redux [43], which are never included in our training dataset. As table 7 shows, although the cost–accuracy trade-off gains are a bit less pronounced on out-of-domain tasks, R2R nonetheless exhibits strong generalization, which continues to outperform the 14B model with less than 7B parameter size.

Table 7: Performance of R2R on Arena-Hard for Dialog and MMLU-Redux-Philosophy.

| Type | Model(s) | Dialog | | | Philosophy | | |
|---|---|---|---|---|---|---|---|
| | | Score | Param. | Cost | Acc. | Param. | Cost |
| LLM | R1-32B | 5.0 | 32 | 65.9 | 79 % | 32 | 32.2 |
| SLM | R1-1.5B | 0.2 | 1.5 | 14.3 | 38 % | 1.5 | 8.7 |
| Distill | R1-7B | 0.3 | 7 | 35.0 | 57 % | 7 | 20.6 |
| Distill | R1-14B | 2.2 | 14 | 56.5 | 77 % | 14 | 18.5 |
| R2R | Ours | **2.8** | **6.1** | **40.9** | **81 %** | **6.7** | **8.3** |

**Generalize to different LLMs**. We test whether a router, trained on data from a specific SLM-LLM pair (e.g., 0.6B-32B), can be effectively applied to a different pair (e.g., 0.6B-8B) without retraining. The experiments are conducted with the Qwen3 model family (0.6B, 8B, and 32B), tested on the AIME (2024+2025) benchmark with a target recall of 0.95 for divergent tokens. The results are summarized in Table 8. Our empirical results indicate acceptable performance when directly substituting the LLM of a trained router without additional retraining. This can be attributed to the common divergence patterns shared by strong LLMs paired with the same weaker SLM.

Table 8: Router generalizability on the AIME benchmark. *Param.* denotes the average activated parameters per token in billions; *Cost* is the average output tokens (thousands) per query multiplied by average parameters (billions) .

| Type | Model(s) | Router Variant | Acc. | Param. | Cost |
|------|----------|----------------|------|--------|------|
| SLM | Qwen3-0.6B | - | 12% | 0.6 | 15 |
| LLM | Qwen3-8B | - | 67% | 8.0 | 130 |
| LLM | Qwen3-32B | - | 75% | 32.0 | 469 |
| R2R | 0.6B+8B | Trained on 0.6B+8B | 65% | 2.7 | 49 |
| | | *Generalized* from 0.6B+32B | 53% | 2.6 | 50 |
| R2R | 0.6B+32B | Trained on 0.6B+32B | 67% | 10.2 | 154 |
| | | *Generalized* from 0.6B+8B | 70% | 9.9 | 151 |

## B.3 Efficiency Analysis

### B.3.1 Data Generation Overhead

Table 9: Latency and GPU usage across different stages of data labeling.

| Stage | Latency (hours) | #GPU | GPU Hour |
|-------|-----------------|------|----------|
| LLM response | 35 | 8 | 280 |
| SLM Prefill | 0.1 | 8 | 0.5 |
| LLM Continuation | 7 | 8 | 56 |
| Verify | 14 | 8 | 112 |
| **Total** | 56 | 8 | 448 |

As Table 9 shows, our four-stage pipeline completes in 56 h (448 GPU hours). The LLM response stage dominates runtime (35 h, 280 GPU hours), but it can be mitigated by directly utilizing the responses of LLMs from open-source SFT datasets, provided they were generated by the same LLM used for routing. The SLM prefill step is highly efficient, requiring only 0.1 h of wall clock time. The subsequent LLM continuation and verification stages take 7 h (56 GPU hours) and 14 h (112 GPU hours), respectively. Compared to downstream tasks, the overall data generation pipeline remains relatively lightweight and efficient.

### B.3.2 Computation and Memory Access

Following prior work [44–46], we analyze the computational and memory access overhead of R2R against various baselines on the AIME benchmark. Table 10 presents a detailed comparison of total computation (TFLOPs), total memory access (TB), and the average memory access per token (GB).

Given that the decoding process in large language models is predominantly memory-bound, the volume of memory access is a critical factor influencing overall throughput. As shown in the table, our method, R2R, markedly reduces total memory access to just 216 TB, a 4.5x reduction compared to the R1-32B model. This efficiency is achieved with only a modest increase in total computation, which primarily arises from the additional KV-cache updates required when switching between the LLM and SLM.

In contrast, speculative decoding approaches such as Eagle2 and HASS incur substantially higher computational costs—more than 22x that of R1-32B. This overhead is a direct consequence of their

Table 10: Computational and memory overhead on the AIME benchmark. R2R demonstrates a significant reduction in memory access compared to the large model (R1-32B) and speculative decoding methods, with only a marginal increase in computation.

| Model | Total Computation (TFLOPs) | Total Memory Access (TB) | Avg. Memory Access Per Token (GB) |
|---|---|---|---|
| R1-1.5B | 157 | 104 | 3.7 |
| R1-14B | 630 | 503 | 29.4 |
| R1-32B | 1136 | 966 | 63.6 |
| Eagle2 | 25490 | 515 | 29.6 |
| HASS | 26809 | 544 | 28.9 |
| R2R | 1502 | 216 | 11.7 |

complex tree-structured draft and verification processes, which can involve up to 60 tokens per cycle. Despite this significant computational demand, these methods also achieve considerable throughput improvements by effectively reducing the memory access bottleneck during the decoding phase. R2R, however, provides a more balanced trade-off, achieving significant memory savings without the extreme computational overhead of speculative methods.

### B.3.3  Inference Latency Breakdown

We integrated R2R into the production-level SGLang serving infrastructure, which is demonstrated by the high absolute throughput results for our method and the baselines. To assess the latency implications of our token-level routing, we performed a detailed runtime breakdown. The analysis was conducted on the AIME benchmark, averaged over 60 questions, using two A800 GPUs.

The results, shown in Table 11, indicate that the token-level routing decisions made by our 56M parameter router introduce minimal latency overhead (5.96%). Given the small size of the neural router, we anticipate that further runtime improvements can be achieved through additional system-level optimizations.

Table 11: Runtime breakdown of the R2R system components on the AIME benchmark.

| Component | Percentage of Total Time |
|---|---|
| Router | 5.96% |
| LLM (R1-32B) | 64.97% |
| SLM (R1-1.5B) | 26.77% |
| Others | 2.30% |

### B.3.4  LLM Activation Intervals

The cost efficiency of the routing system largely depends on how frequently the LLM is activated during generation. Since the *EXTEND* operation of the LLM is inherently compute-bounded when consecutive LLM calls are spaced further apart. In other words, given a fixed LLM activation rate, longer intervals between two LLM invocations lead to better GPU utilization and less scheduling overhead per generated token.

To better understand this effect, we analyze the distribution of the interval between consecutive divergent tokens. Figure 7 illustrates these distributions for both the training dataset and actual inference traces on AIME25. As shown, the actual distribution closely aligns with that observed in training dataset, suggesting that the router generalizes well to real inference scenarios. More importantly, both distributions exhibit a long-tailed shape, indicating that divergence events occur sparsely and irregularly, which allows R2R to sustain high computational efficiency without overusing the LLM.

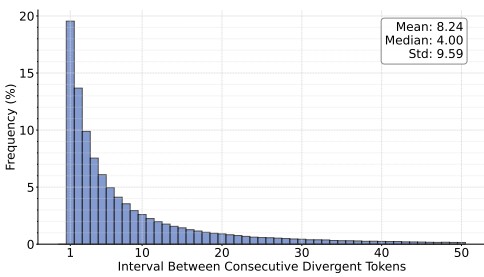 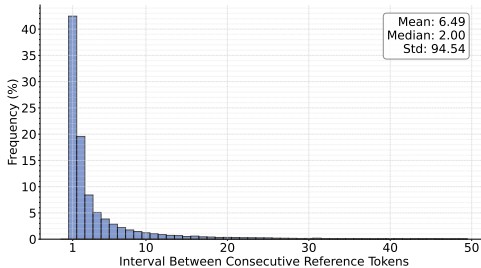

Figure 7: (a) Divergent token interval distribution for training dataset; (b) Reference token interval distribution on AIME25

## B.4 Comparison and Compatibility with Orthogonal Techniques

### B.4.1 R2R and Mixture-of-Experts (MoE)

We compare our method, R2R, with the Mixture-of-Experts (MoE) architecture, another prominent approach for efficient model scaling. This section begins with a conceptual comparison, followed by empirical results, and concludes by discussing how the two methods can be integrated to create a more favorable performance-efficiency trade-off.

**Conceptual comparison**. As noted, both MoE and R2R leverage partial activation, but they differ in key design aspects, as summarized in Table 12. This conceptual distinction motivates their potential integration, which we explore next.

Table 12: Conceptual comparison between MoE and R2R.

| Design Aspect | MoE | R2R |
|---|---|---|
| Partial activation | Yes | Yes |
| Routing granularity | Fine-grained (Experts) | Coarse-grained (Models) |
| Subject model sizes | Equal parameters per expert | Different parameters per model |
| Training overhead | Full training from scratch | Router training only |
| Supervision | Next-token predictions | Explicit routing decisions |

**Empirical comparison**. We evaluate the MoE model Qwen3-30BA3B and an R2R model (routing between Qwen3-0.6B and Qwen3-8B) on the AIME benchmark. The 60 questions are split into Easy (28) and Hard (32) subsets based on whether the question ID is less than or equal to 7. The results in Table 13 highlight the complementary strengths of each approach. Thanks to extensive pretraining, MoE models can match dense model performance on challenging tasks. However, their fixed per-token overhead limits efficiency on simpler inputs. In contrast, R2R dynamically adjusts computation based on token-level *divergence*, achieving lower cost on easier examples, though its lighter training cannot match MoE performance on harder questions given the same activated parameters. Given their complementary advantages, we explore the integration of both methods.

Table 13: Empirical comparison of R2R and MoE on the AIME benchmark, split by difficulty.

| Type | Model(s) | Easy | | Hard | |
|---|---|---|---|---|---|
| | | Acc. | Param. | Acc. | Param. |
| R2R | Qwen3-0.6B + Qwen3-8B | 93% | 2.9 | 41% | 2.6 |
| MoE | Qwen3-30BA3B | 93% | 3.3 | 59% | 3.3 |

**MoE for R2R**. R2R is directly compatible with efficiency-optimized models like MoEs. We validate R2R's performance using the Qwen3-30BA3B MoE as the LLM, as shown in Table 14. Combining R2R with an MoE model significantly improves the Pareto frontier, achieving robust accuracy at extremely low average activated parameters.

Table 14: Performance of R2R when using an MoE model as the LLM.

| Type | Model | Acc. | Param. | Cost |
|------|-------|------|--------|------|
| Dense | Qwen3-0.6B | 12% | 0.6 | 15 |
| Dense | Qwen3-1.7B | 37% | 1.7 | 33 |
| Dense | Qwen3-32B | 75% | 32.0 | 469 |
| **R2R** | **0.6B + 32B** | **67%** | **10.2** | **154** |
| MoE | Qwen3-30BA3B | 75% | 3.0 | 51 |
| **R2R** | **0.6B + 30BA3B** | **68%** | **1.3** | **22** |

**R2R for MoE**. Conversely, R2R's principles can enhance future MoE designs. Current MoE methods generally use uniform-sized experts; introducing mixed-sized experts could enable query-adaptive activation. Furthermore, supplementing an MoE's training objective with explicit *divergence* or difficulty-based routing supervision from R2R may encourage more adaptive usage of larger experts for only the most critical decoding steps. Although retraining MoE models is beyond this paper's scope, this presents an exciting direction for future research.

### B.4.2 R2R and Query-level Routing (QR)

Query-level routing (QR) is methodologically orthogonal to R2R. Consequently, the composition of the two is a natural and well-motivated design choice. We investigated combining R2R with query-level routing methods using the Qwen3 model series with varied sizes, evaluated on the AIME benchmark. Specifically, we treated R2R-S (0.6B+8B) and R2R-L (0.6B+32B) as the SLM and LLM, respectively, for query-level routing.

Table 15: Performance of R2R when combining R2R and query-level routing (QR).

| Type | Model | Acc. | Param. (B) | Cost (KB) |
|------|-------|------|-----------|-----------|
| SLM | Qwen3-0.6B | 12% | 0.6 | 15 |
| LLM | Qwen3-8B | 67% | 8 | 130 |
| LLM | Qwen3-32B | 75% | 32 | 469 |
| R2R-S | Qwen3-0.6B + 8B | 59% | 2.1 | 40 |
| R2R-L | Qwen3-0.6B + 32B | 67% | 10.2 | 154 |
| R2R-M | Qwen3-4B + 8B | 65% | 5.4 | 95 |
| QR-BERT | Qwen3-0.6B + 32B | 25% | 5.4 | 230 |
| QR-LLM | Qwen3-0.6B + 32B | 23% | 5.4 | 238 |
| QR-MF | Qwen3-0.6B + 32B | 32% | 5.4 | 216 |
| QR-SW | Qwen3-0.6B + 32B | 30% | 5.4 | 220 |
| QR-BERT | R2R-S + R2R-L | 63% | 5.4 | 179 |
| **QR-LLM** | **R2R-S + R2R-L** | **67%** | **5.4** | **175** |
| QR-MF | R2R-S + R2R-L | 58% | 5.4 | 181 |
| QR-SW | R2R-S + R2R-L | 62% | 5.4 | 183 |

From Table 15, we can conclude that (1) R2R models generally serve as superior SLM/LLM candidates for query-level routing compared to original single-model setups, consistently advancing the Pareto frontier; (2) The interaction between medium-sized R2R models and query-level routing involving smaller or larger R2R-LLMs is empirically non-trivial. This observation suggests exciting opportunities for future research into sophisticated routing strategies, which combine the search space of query and token-level routing methods.

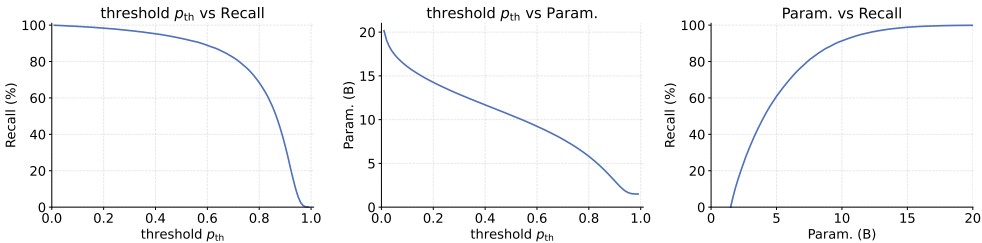

Figure 8: Relationship between the routing threshold, recall for divergent tokens, and average parameter size. The average parameter size is computed based on the positive prediction rate of the router at each threshold.

## B.5   Additional Discussions and Analysis

### B.5.1   Influence of Routing Threshold

Figure 8 visualizes how the routing threshold $p_{th}$ affects the average parameter size on the validation dataset. In our experiments, we select thresholds based on the user-specified average parameter budget (e.g., 6B) measured on the validation set. However, the threshold does not strictly guarantee the same average parameter size during inference, particularly when query difficulty varies significantly from the validation set. Empirically, we observe minimal variance between the target and actual parameter sizes, as the difficulty of our validation and evaluation datasets is generally well aligned. For tasks that are much easier or harder than those considered in this paper, users can construct a tailored validation set using our data labeling pipeline to determine an appropriate threshold.

Beyond analyzing the relationship between threshold and average parameter size, we also examine its effect on divergent token recall. This links the average parameter size to the ability to recall divergent tokens. As shown in Figure 8, recall rises rapidly with increasing average parameter size, demonstrating the strong predictive performance of our router model.

### B.5.2   Sentence-level Path-following Routing

The sentence-level path-following assumption, while necessary to manage the vast search space, could be limiting in certain scenarios. For instance, an SLM might rearrange sentences in a paragraph while still preserving a valid reasoning path, yet our sentence-level verifier might incorrectly flag this as a divergence. While such mislabeling of a *neutral* continuation as *divergent* does not degrade final performance, it can introduce unnecessary computational overhead by triggering the LLM more often than required.

To empirically evaluate the impact of this assumption, we extended our routing strategy to a more general *N-sentence path-following* approach. In this setup, the continuation-and-verification process described in Section 3.2 is not stopped by the first sentence-ending token, but continues for N sentences. As N increases, the strategy transitions smoothly from our default sentence-level approach (N=1) towards the full path-following routing defined in Section 3.1, which is guaranteed to find the optimal path (see Appendix E).

We conducted experiments on a subset of the AIME dataset to measure the effect of varying N. We found that increasing N from 1 to 5 only yields a modest reduction in the measured divergence rate, from 2.91% to 2.66%. This corresponds to a marginal decrease of 0.08B in the average activated parameters required to maintain the same level of accuracy.

Given the limited performance gain relative to the substantial increase in data labeling complexity and cost, we empirically selected N=1 (sentence-level routing) as the most practical and efficient default setting for our framework.

### B.5.3 Ablation Study Results

To further investigate the impact of different routing strategies, we perform an ablation study that analyzes the relationship between *Positive Rate*, defined as the fraction of tokens routed to the LLM, and *Recall*, defined as the proportion of correctly identified positive samples in the training dataset.

**Routing objective**. As illustrated in Figure 9, R2R (default) consistently achieves higher recall across all positive prediction rates compared to the other variants, demonstrating that its routing policy is more accurate in identifying informative tokens. The *different* variant, which substitutes the semantic divergence metric with a simpler criterion that only considers mismatched predictions, exhibits a noticeable drop in recall. This degradation highlights the importance of fine-grained semantic alignment in accurately distinguishing divergence tokens and maintaining routing precision.

**Routing input**. Figure 9 also shows that *HS* and *HS+Token* variants, which rely solely on hidden-state distance or only combine it with current token ids, yield similar but lower recall curves, indicating that low-level representation similarity alone is insufficient to capture reasoning divergence and prove the significance of the logits during routing. In contrast, R2R achieves both higher efficiency and better coverage, maintaining near-optimal recall with less than 14% of tokens routed to the LLM.

**SLM-LLM combination**. To analyze the performance of R2R under difference model pairs, we apply it to the Qwen3 family with a fixed LLM selection (Qwen3-8B) and vary the SLM to form three different SLM-LLM combinations: Qwen3-0.6B+8B, Qwen3-1.7B+8B, and Qwen3-4B+8B. All combinations were tested on AIME with identical settings. Figure 10 shows the relationship between accuracy and the average activated parameters per token: For a fixed LLM, smaller SLMs yield a better Pareto frontier, matching accuracy at lower cost or achieving higher accuracy at the same cost. The intuition is that, for identical tokens that can be predicted by multiple SLMs, routing them to the smaller SLM can preserve correctness of the prediction with less computational cost, thereby improving the overall accuracy–efficiency trade-off.

Interestingly, we found that all seven AIME questions that the SLM answered correctly were also answered correctly by the LLM, indicating that the LLM consistently outperforms the SLM rather than specializing in a different subset of math problems.

These results validate that the semantic divergence criterion used in R2R provides a more discriminative signal for routing, effectively balancing accuracy and efficiency in the overall system. Moreover, appropriate selection of the SLM–LLM combination can further improve the cost–accuracy Pareto frontier.

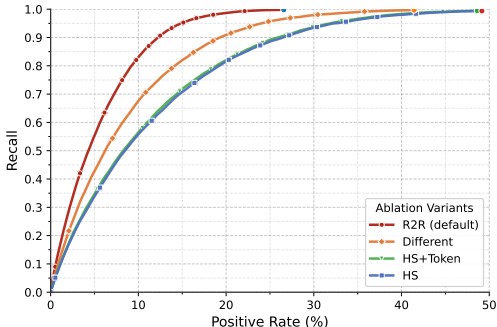

Figure 9: Relationship between the ratio of LLM usage and the recall of divergent tokens in the validation dataset.

Figure 10: Scaling of accuracy versus activated parameters per token, evaluated on AIME across different model combinations in Qwen3 family

### B.6 Systematic Failure Analysis

To identify systematic failure modes, we analyzed the response status of R2R (routing between R1-1.5B and R1-32B without sampling) on the AIME and GPQA benchmarks. As shown in Table 16, the primary failure mode occurs when R2R cannot complete its reasoning within the 32K token limit. This typically happens due to repetitive reasoning patterns that were not encountered during the router's training phase. Additionally, we observe that R2R tends to invoke the LLM slightly

more frequently on the GPQA benchmark. This is particularly noticeable for queries involving rare tokens, such as complex protein or chemical compound names, where the SLM's uncertainty is higher, triggering the router.

Table 16: Systematic failure analysis of R2R on AIME and GPQA benchmarks.

| Benchmark | Correct | Unfinished | Wrong (Same as) | | Wrong (Diff. from) | |
|---|---|---|---|---|---|---|
| | | | LLM | SLM | LLM | SLM |
| AIME (60) | 33 | 25 | 0 | 0 | 2 | 2 |
| GPQA (198) | 87 | 85 | 6 | 3 | 20 | 23 |
| Total (258) | 120 | 110 | 6 | 3 | 22 | 25 |

## B.7 Performance-Cost Trade-off

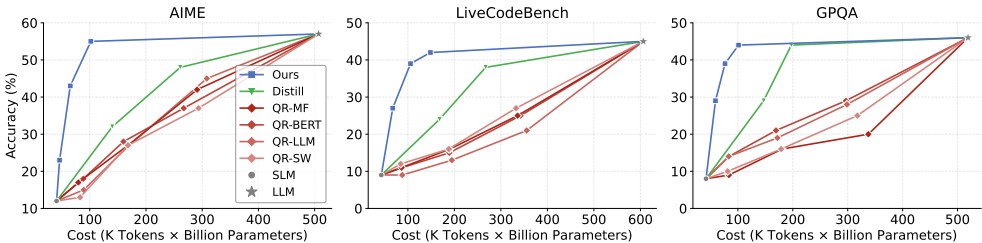

Figure 11: Scaling of accuracy versus total cost, evaluated across AIME, GPQA, and LiveCodeBench. R2R advances the Pareto frontier beyond distillation and query-level routing methods.

Figure 11 illustrates the trade-off between accuracy and the total cost of generation. As defined in Section 5.1, the cost metric is calculated as the average number of output tokens multiplied by the average parameter size, serving as a hardware-agnostic indicator of latency across methods. R2R consistently outperforms both query-level routing methods and distilled LLMs, establishing a new Pareto frontier in performance-cost trade-offs.

# C  Additional Design Discussions

## C.1  Extend R2R to More Sampling Methods

### C.1.1  Path-Following Routing Under Sampling

R2R naturally extends to sampling-based decoding by adapting *divergence* labeling to a probabilistic setting. Specifically, the deterministic continuation sequence pair $(S_s, S_l)$ from Equations 3–5 are replaced by $k$ sampled continuation pairs under a chosen sampling method. *Divergence* is then evaluated for each sampled pair. The routing decision for each token then depends on whether the overall divergence probability exceeds a predefined threshold:

$$P(V(S_{s_i}, S_{l_i}) = 0 | i \in [0, k)) \geq P_{\text{threshold}}$$

If setting $P_{\text{threshold}}$ to the LLM's self-*divergence* probability $P(V(S_l, S_l) = 0)$, it ensures the mixed model, under full path-following routing, maintains the same quality expectation as the LLM alone.

### C.1.2  Efficient Data Generation

Due to sampling stochasticity, each mismatched token can branch into multiple continuations. However, sampling multiple continuations for *divergence* labeling is computationally expensive. Since continuations are sentence-level, we empirically observe that the *divergence* decision primarily depends on the first differing token between the SLM and LLM samples, rather than the subsequent

continuations. To efficiently approximate probabilistic routing, we therefore only sample for next-token generation $y_i(\theta_s|S_{<i})$ and $y_i(\theta_l|S_{<i})$, while leaving continuations deterministic.

Given the extensive volume of tokens, we set $k = 1$ and $P_{\text{threshold}} = 0.5$, simplifying the setup. This approximation incurs overhead comparable to our greedy data generation pipeline. In implementation, our data-generation pipeline adapts to the new setup by using sampling-based generation for both LLM responses (step 0) and SLM prefill (step 1), while keeping other procedures unchanged.

Table 17: Statistics of tokens difference and divergence across query types in the **sampling-based** training dataset.

| Type | #Query | #Token | #Different | Diff. Rate | #Divergent | Div. Rate |
|------|--------|--------|------------|------------|------------|-----------|
| Math | 862 | 7.3M | 858K | 11.8% | 438K | 6.0% |
| Code | 987 | 6.8M | 1.2M | 17.5% | 627K | 9.3% |
| QA | 805 | 2.1M | 443K | 21.3% | 231K | 11.1% |
| Summary | 2654 | 16.1M | 2.5M | 15.5% | 1.3M | 8.0% |

Using this modified pipeline, we generate 16.1 million routing labels, covering topics of math, coding, and QA with queries from the Bespoke-Stratos dataset. Table 17 summarizes the statistics of the generated training dataset with the sampling method. Compared with the non-sampled dataset in Table 1, sampling increases the average divergence rate by only 3.1%, which remains low overall, indicating divergence patterns consistent with greedy decoding.

### C.1.3 Experiment Results

For preliminary experiments, we use DeepSeek-R1's recommended sampling settings (temperature = 0.6, top-p = 0.95) for both dataset generation and evaluation. The neural router remains unchanged, as it already takes the sampled token's embedding as input.

We evaluate the extended R2R on the AIME benchmark, reporting its **pass@1** accuracy over 16 independent samples per problem, and compare it against query-level routing baselines following Section 5.1. As shown in Table 18, R2R continues to advance the Pareto frontier, achieving 1.4–1.5$\times$ higher AIME scores than query-level routing methods and matching the R1-14B performance with only 8.6B average activated parameters. Extending R2R to stochastic decoding preserves its Pareto optimality relative to both distilled models and query-level routing approaches.

Table 18: Performance and efficiency comparison on AIME and methods with sampling. *Param.* denotes the average activated parameters per token in billions; *Cost* is the average output tokens (thousands) per query multiplied by average parameters (billions) .

| Method | Acc. | Param. (B) | Cost (KB) |
|--------|------|------------|-----------|
| R1-1.5B | 27% | 1.5 | 24.3 |
| R1-32B | 61% | 32.0 | 384.8 |
| R1-14B | 58% | 14.0 | 170.5 |
| QR-SW | 39% | 9.2 | 135.5 |
| QR-LLM | 38% | 9.2 | 138.6 |
| QR-BERT | 40% | 9.4 | 136.8 |
| QR-MF | 41% | 9.1 | 130.5 |
| R2R | **58%** | **8.6** | **115.3** |

### C.2 Routing Algorithm Details

As illustrated in Algorithm 1, the objective of our routing algorithm is to identify and correct path-divergent tokens during inference by using predictions from a large language model (LLM). Both SLM and LLM perform greedy decoding, and a token is considered identical if both models produce the same prediction.

When the SLM and LLM outputs differ, the algorithm triggers a continuation process for each model: the SLM and LLM, respectively, continue generating tokens, starting from their initial divergent prediction, until a special separator (SEP) token is produced. These continuations yield two complete sequences that differ only at the initial divergence point and subsequent tokens.

To assess whether this divergence impacts reasoning, a separate LLM-based verifier is employed. This verifier receives the two generated sequences and outputs a binary decision: 0 if the sequences are semantically neutral, and 1 if they diverge significantly in meaning or logic.

If the verifier outputs 0 (neutral), the router accepts the SLM's prediction. However, if the verifier outputs 1 (divergent), the algorithm corrects the current token by adopting the LLM's prediction, thus preventing further drift from the intended reasoning path.

This approach ensures that the system maintains high alignment with LLM reasoning, while minimizing unnecessary reliance on the LLM by routing to the more efficient SLM whenever possible.

---

**Algorithm 1** Path-Following Routing

**Input:** Partial sequence $S_{<i}$, models $\{\theta_s, \theta_l\}$
**Output:** Selected model $m_i$
1: $y_s \leftarrow \arg\max_y P_{\theta_s}(y \mid S_{<i})$
2: $y_l \leftarrow \arg\max_y P_{\theta_l}(y \mid S_{<i})$
3: **if** $y_s = y_l$ **then**
4: $\quad m_i \leftarrow \theta_s$ $\qquad\qquad\triangleright$ *identical*
5: **else**
6: $\quad S_s \leftarrow$ CONTINUATION$(S_{<i}, y_s)$
7: $\quad S_l \leftarrow$ CONTINUATION$(S_{<i}, y_l)$
8: $\quad$ **if** $J_e(S_s, S_l) = 0$ **then**
9: $\qquad m_i \leftarrow \theta_s$ $\qquad\qquad\triangleright$ *neutral*
10: $\quad$ **else**
11: $\qquad m_i \leftarrow \theta_l$ $\qquad\qquad\triangleright$ *divergent*
12: $\quad$ **end if**
13: **end if**
14: **return** $m_i$

---

**Algorithm 2** Continuation $(S, y)$

**Input:** Prefix sequence $S$, initial token $y$
**Output:** Completed sequence $S$
1: $S \leftarrow S + y$
2: **while** $y \notin$ SEP **do**
3: $\quad y \leftarrow \arg\max_{y'} P_{\theta_l}(y' \mid S)$
4: $\quad S \leftarrow S + y$
5: **end while**
6: **return** $S$

---

**Algorithm 3** LLM Verifier $\mathcal{V}(S_s, S_l)$

**Input:** Sequences $S_s, S_l$
**Output:** $o$ (0: neutral, 1: divergent)
1: $o \leftarrow$ LLM verifies if $S_s$ and $S_l$ diverges
2: **return** $o$

---

# D   Additional Related Work

## D.1   Model-level Compression

Extensive studies have been proposed to accelerate the costly decoding processes of LLM by compressing the models themselves [47]. Prominent techniques include sparse attention mechanisms [48–52] and model quantization [53–58]. In contrast, our R2R method focuses on optimizing inference *above* the model level, complementing these model compression techniques. Therefore, it can be effectively combined with them to further enhance inference efficiency.

## D.2   Concurrent Mix Inference Methods

Given recent rapid advancements in reasoning LLMs, several concurrent studies also explore mix inference strategies that integrate small and large language models. These methods differ primarily in their routing granularity, objectives, and specific routing schemes.

**Step-level Methods**: Speculative Thinking [59], SplitReason [35], and SpecReason [60] operate at the reasoning step granularity. Speculative Thinking observes that *LLMs excel at affirmation and reflection compared to SLMs*. Thus, it employs heuristic triggers—such as affirmations ("yeah"), reflections ("wait"), or verification signals ("check")—to invoke the LLM selectively after detecting delimiter tokens (e.g., "\n\n"), enhancing subsequent SLM-generated segments. SplitReason aims to *offload difficult reasoning steps to the LLM*. It first uses a strong LLM to identify challenging reasoning steps, then trains the SLM to generate a special token (i.e., '<bigmodel>') signaling the LLM to take over these difficult steps. SpecReason *adapts speculative decoding to reasoning*

*step-level*. It utilizes the LLM to evaluate steps generated by the SLM, reverting to the LLM only when the score of SLM-generated steps falls below a certain threshold.

**Token-level Methods**: Unlike step-level methods, CITER [61] and RSD [62] adopt finer-grained token-level routing strategies. CITER formulates the routing problem as a long-horizon reinforcement learning task, optimizing for final answer quality and inference efficiency. Because CITER targets general decoding tasks (e.g., short-form QA), repeatedly generating the complete response to determine token-level preferences remains computationally manageable. In contrast, RSD leverages an existing reward model to dynamically select tokens for LLM generation whenever the SLM-produced tokens exhibit low reward scores. This approach performs well on tasks with clear and definable reward signals.

**Distinctiveness of R2R**: R2R distinguishes itself from concurrent works by specifically targeting *immediate divergence correction at token granularity*. Unlike methods focused on offloading complex reasoning steps, R2R addresses the subtle scenario where the SLM and LLM may agree on challenging steps, yet diverge unexpectedly on seemingly straightforward tokens (under human or LLM judgment). Such divergences can significantly alter the subsequent reasoning path, thus requiring immediate correction. Moreover, R2R differs from the speculative decoding scheme, as it does not rely on periodic LLM verification steps to inform routing decisions. Instead, R2R immediately routes divergent tokens to the LLM, effectively preventing divergence without incurring rollback overhead.

Given these distinct objectives and design choices, integrating R2R with these concurrent methods represents a promising direction for future research, enabling even more effective mix inference frameworks.

# E  Proof of Quality Guarantee for Full Path-Following Routing

## E.1  Notations

We summarize the notations used throughout this proof:

- $\theta_l$, $\theta_s$: the large and small language models (LLM, SLM), respectively.
- $S_{<i} = [x_0, \ldots, x_{n-1}, y_0, \ldots, y_{i-1}]$: the prefix sequence up to, but not including, position $i$, where $S_{<0}$ contains only the input tokens.
- $y_i(m|S_{<i})$: the token generated at position $i$ by model $m \in \{\theta_s, \theta_l\}$ given prefix $S_{<i}$.
- $\mathcal{V}(\cdot, \cdot) \in \{0, 1\}$: the quality verifier function, returning 1 iff the first sequence achieves the same quality as the second.
- $S_{<i}^{(L)}$: sequence up to $i$ generated by the LLM only.
- $S_{<i}^{(M)}$: sequence up to $i$ generated by the mixed (routed) strategy.
- $\mathcal{S}_s, \mathcal{S}_l$: continuation sequences as defined in Equations 8 and 9.
- $\mathcal{T}_{<i}$: the sequence formed by $S_{<i}^{(M)}$ concatenated with the LLM's continuation tokens:

$$\mathcal{T}_{<i} = S_{<i}^{(M)} \oplus [y_i(\theta_l|S_{<i}^{(M)}), y_{i+1}(\theta_l|S_{<i}^{(M)} \oplus y_i(\theta_l|S_{<i}^{(M)})), \ldots, \text{EOS}] \qquad (6)$$

The routing decision at each step $i$ is given by:

$$m_i = \begin{cases} \theta_s, & \underbrace{y_i(\theta_s|S_{<i}) = y_i(\theta_l|S_{<i})}_{identical} \quad \text{or} \quad \underbrace{\mathcal{V}(\mathcal{S}_s, \mathcal{S}_l) = 1}_{neutral} \\ \theta_l, & \underbrace{\mathcal{V}(\mathcal{S}_s, \mathcal{S}_l) = 0}_{divergent} \end{cases} \qquad (7)$$

The continuation sequences after the $i$-th token are:

$$\mathcal{S}_s = S_{<i} \oplus [y_i(\theta_s|S_{<i})] \oplus [y_{i+1}(\theta_l|S_{<i} \oplus y_i(\theta_s|S_{<i})), \ldots, \text{EOS}] \qquad (8)$$

$$\mathcal{S}_l = S_{<i} \oplus [y_i(\theta_l|S_{<i})] \oplus [y_{i+1}(\theta_l|S_{<i} \oplus y_i(\theta_l|S_{<i})), \ldots, \text{EOS}] \qquad (9)$$

where $\oplus$ denotes sequence concatenation.

## E.2 Theorem

The sequence $S^{(M)}$, generated by the path-following routing strategy, is guaranteed to achieve the same quality as its LLM-only counterpart $S^{(L)}$ under $\mathcal{V}$.

## E.3 Proof

**Base Case.** At $i = 0$, $S_{<0}^{(M)} = S_{<0}^{(L)} = [x_0, \ldots, x_{n-1}]$, so $\mathcal{T}_{<0}$ is simply the LLM's full sequence. Thus, $\mathcal{V}(\mathcal{T}_{<0}, S^{(L)}) = 1$.

**Inductive Hypothesis.** Suppose that for some $k$ with $0 \leq k < n$, we have $\mathcal{V}(\mathcal{T}_{<k}, S^{(L)}) = 1$; i.e., continuing from $S_{<k}^{(M)}$ with the LLM produces a sequence of equal quality to $S^{(L)}$.

**Inductive Step.** Consider the $(k+1)$-th token $y_k^{(M)}$ determined by Eq. 7. There are three cases:

    (a) **Identical:** $y_k(\theta_s|S_{<k}^{(M)}) = y_k(\theta_l|S_{<k}^{(M)})$. Then $y_k^{(M)}$ matches the LLM's output, so the sequence remains identical and $\mathcal{V}(\mathcal{T}_{<k+1}, S^{(L)}) = 1$.

    (b) **Neutral:** $y_k(\theta_s|S_{<k}^{(M)}) \neq y_k(\theta_l|S_{<k}^{(M)})$ but $\mathcal{V}(\mathcal{S}_s, \mathcal{S}_l) = 1$. The router selects the SLM token, so $\mathcal{T}_{<k+1} = \mathcal{S}_s$. By definition, $\mathcal{V}(\mathcal{T}_{<k+1}, S^{(L)}) = 1$.

    (c) **Divergent:** $\mathcal{V}(\mathcal{S}_s, \mathcal{S}_l) = 0$. The router selects the LLM token, so the mixed sequence again matches the LLM's output, and thus $\mathcal{V}(\mathcal{T}_{<k+1}, S^{(L)}) = 1$.

**Conclusion.** By mathematical induction, for all $i \in [0, n]$, the continuation $\mathcal{T}_{<i}$ maintains the same quality as $S^{(L)}$ under $\mathcal{V}$. At generation completion $(i = n)$, $S^{(M)} = S_{<n+1}^{(M)}$, so $\mathcal{V}(S^{(M)}, S^{(L)}) = 1$.

∎

# F Prompts and Examples

## F.1 Prompt for Verifier Model

As discussed in Section 4.1, we design a structured prompt for the verifier model (Qwen2.5-72B-Instruct) to assess whether divergences between two sentences affect their meaning, reasoning, logic, or conclusions. Please refer to Text 2. for the exact prompt. The prompt highlights the divergence point and provides explicit criteria for labeling. It instructs the model to justify its judgment with a brief explanation and includes illustrative examples to guide the model's understanding of both scenarios.

## F.2 Response Example

We use an example question from AIME and responses from R1-1.5B, R1-32B and R2R to provide an intuitive understanding of our method.

> **Text 1. Question**
>
> Find the largest possible real part of
> $$(75 + 117i)z + \frac{96 + 144i}{z}$$
> where $z$ is a complex number with $|z| = 4$.

Text 3-5 shows the example responses. The R1-1.5B and R1-32B models produce distinct final answers for the maximum real part, reflecting a divergence in their reasoning paths. By contrast, R2R identifies and corrects this divergence, navigating the correct reasoning path to get the final answer matches that of the stronger 32B model. At the same time, R2R tolerates neutral differences—such

as minor phrasing or presentation—between models when these do not affect the core reasoning or conclusions. This selective routing mechanism enables R2R to deliver both high efficiency and accuracy by only invoking the large model for tokens that would otherwise lead to substantive differences in meaning or logic.

## Text 2. Prompt For Verifier Model

**Task:**
Determine if the divergence between Sentence 1 and Sentence 2 affects the meaning, reasoning, logic, or conclusions derived from them.

**Instructions:**
- The marker « » indicates where the sentences diverge. It is **not** part of the original text.
- Assess whether this divergence changes the meaning, reasoning, logic, or conclusions, or if it introduces new information or contradictions.

**Output 1 if:**
- The divergence causes a change in meaning, reasoning, logic, or conclusions.
- It introduces new information, shifts focus, or contradicts prior facts.
- The sentences follow different reasoning paths or focus on different aspects.

**Output 0 if:**
- The divergence is superficial and does not affect meaning, reasoning, logic, or conclusions.
- Both sentences follow the same reasoning path or lead to the same conclusion.

**Reasoning:** Provide a brief explanation of how the divergence impacts (or does not impact) meaning, reasoning, logic, or conclusions.

—

**Example 1 (Same - 0):**
Sentence 1:
`"The ratio of adults to total people «is» now 11/25."`
Sentence 2:
`"The ratio of adults to total people «chang»ed from 5/12 to 11/25`
`after adding 50 people."`
**Output:** 0
**Reasoning:** The change from "is" to "changed" does not affect the overall meaning, reasoning, logic, or conclusions.

**Example 2 (Different - 1):**
Sentence 1:
`"Let's solve this using «integration» by parts."`
Sentence 2:
`"Let's solve this using «u-»substitution."`
**Output:** 1
**Reasoning:** The change in method (from integration by parts to substitution) alters the reasoning and approach to solving the problem.

—

**Now complete the task:**
**Common Context:**
`{CONTEXT}`
Sentence 1:
`{SENTENCE1}`
Sentence 2:
`{SENTENCE2}`
**Answer (Output: <0 or 1>)**
**Reasoning:**

## Text 3. **R1-1.5B Response**

Okay, so I have this problem: I need to find the largest possible real part of the expression

$$(75+117i)z+\frac{96+144i}{z}$$

, where $z$ is a complex number with $|z| = 4$. Hmm, complex numbers can sometimes be tricky, but let me try to break this down step by step.

First, I know that any complex number $z$ can be written in polar form as $z = 4e^{i\theta}$ since its magnitude is 4. That might make things easier because then I can express $z$ in terms of its angle $\theta$. So, let me write that down:

$z = 4e^{i\theta}$

Then, $\frac{1}{z}$ would be $\frac{1}{4}e^{-i\theta}$ because the reciprocal of $e^{i\theta}$ is $e^{-i\theta}$.

So, substituting back into the expression, we have:

$(75 + 117i) \cdot 4e^{i\theta} + \frac{96+144i}{4e^{i\theta}}$

Let me compute each term separately. First, let's compute $(75 + 117i) \cdot 4e^{i\theta}$.

Multiplying $75 + 117i$ by 4 gives:

$4 \times 75 = 300$ and $4 \times 117i = 468i$, so the first term is $300 + 468i$.
. . .

The final answer is:

$$\boxed{300 + 12\sqrt{13}}$$

## Text 4. **R1-32B Response**

Okay, so I have this problem here: I need to find the largest possible real part of the expression

$$(75+117i)z+\frac{96+144i}{z}$$

where $z$ is a complex number with $|z| = 4$. Hmm, that sounds a bit tricky, but let me try to break it down step by step.

First, I know that $z$ is a complex number with magnitude 4. So, I can represent $z$ in polar form. That might make things easier because working with polar forms can simplify multiplication and division. So, let me write $z = 4e^{i\theta}$, where $\theta$ is the argument of $z$. That way, $|z| = 4$ is automatically satisfied.

Now, let's substitute $z = 4e^{i\theta}$ into the expression. So, the expression becomes:

$(75 + 117i) \cdot 4e^{i\theta} + \frac{96+144i}{4e^{i\theta}}$

Simplify each term separately. Let's start with the first term:

$$(75 + 117i) \cdot 4e^{i\theta}$$
$$=4(75 + 117i)e^{i\theta}$$

. . .

$4 \times 75\cos\theta = 300\cos\theta$
$4 \times (-117\sin\theta) = -468\sin\theta$
. . .

Thus, the largest possible real part is:

$$\boxed{540}$$

## Text 5. **R2R Response**

Okay, so I have this problem: I need to find the largest possible real part of the expression

$$(75+117i)z+\frac{96+144i}{z}$$

, where $z$ is a complex number with $|z| = 4$. Hmm, that sounds a bit tricky, but let me try to break it down step by step.

First, I know that $z$ is a complex number with magnitude 4. So, I can represent $z$ in polar form as $z = 4e^{i\theta}$, where $\theta$ is the argument of $z$. That might make things easier because working with exponentials can sometimes simplify multiplication and division. So, let me rewrite the expression using this polar form. The expression is:

$$(75+117i)z+\frac{96+144i}{z}$$

Substituting $z = 4e^{i\theta}$, we get:

$(75 + 117i)(4e^{i\theta}) + \frac{96+144i}{4e^{i\theta}}$

Simplify each term:
First term:

$$(75 + 117i) \cdot 4e^{i\theta}$$
$$=4(75 + 117i)e^{i\theta}$$

. . .

Which is:
$300\cos\theta - 468\sin\theta + i(300\sin\theta + 468\cos\theta)$
. . .

Thus, the largest possible real part is:

$$\boxed{540}$$

