# OpenReview forum: "R2R: Efficiently Navigating Divergent Reasoning Paths with Small-Large Model Token Routing"
_NeurIPS.cc/2025/Conference — NeurIPS 2025 poster_

### Official Review · Reviewer_gjwQ · 2025-06-29

**Clarity:** 3
**Significance:** 3
**Originality:** 3
**Rating:** 5
**Confidence:** 3

**Summary:**

This paper proposes a token-level routing method to decide whether to use a small language model (SLM) or a large language model (LLM) for each token. The goal is to have the SLM follow the LLM's reasoning path and it seems that only a small fraction of tokens diverge the reasoning paths. Therefore, by identifying those divergent tokens and using the LLM for generating those while using the SLM for other tokens can lead to comparable results with the LLM while having smaller number of activated parameters and having a significant speedup compared to just using the LLM for the entire generation. Their main contributions are (1) proposing an automatic pipeline to label divergent tokens. (2) developing a token-level routing scheme using a lightweight neural router. (3) enabling a more efficient performance scaling at test time using their R2R framework.

**Questions:**

(1) What are your thoughts on combining R2R with some query-level routing methods so that you can choose which SLM or LLM to use based on the difficulty of the query, and then using R2R to route tokens to the SLM or LLM for the generation?

(2) Can you please clarify what you mean in lines 170-171? I assumed you would rely on the smaller model for generating the majority of the tokens. Why is it said that only 3% of generated tokens are from the smaller model?

**Ethical Concerns:**

["NO or VERY MINOR ethics concerns only"]

**Final Justification:**

The authors have provided clear and thorough explanations in their rebuttal, and I will maintain my positive assessment of this paper.

**Limitations:**

Their method is based on greedy decoding and sentence-level assumptions. These assumptions can be limiting. For example, the SLM might want to generate the sentences in a paragraph in a different order but still follow a valid reasoning path and achieve the same conclusion as the LLM in some cases. Also, the greedy decoding may potentially be problematic for tasks where we want to generate more diverse or creative text.

**Paper Formatting Concerns:**

I didn't notice any major formatting issues.

**Quality:**

3

**Strengths And Weaknesses:**

Strengths:

(1) This work introduces a token-level approach that enables small language models to follow the reasoning paths of a larger model by identifying the divergent tokens and only correcting those by using the LLM. This allows the SLM to achieve comparable performance with the LLM while using the LLM only for generating a small fraction of tokens. This is significant as this can achieve a more efficient performance scaling at test time.

(2) The lightweight routing scheme identifies whether to use the LLM or SLM at each token level, and it can immediately decide whether to accept the SLM generation or not. Therefore, it eliminates the need for rollbacks, which methods like speculative decoding usually face (Speculative decoding methods only periodically verify the tokens and invalidate all tokens after a correction).

(3) The authors report the results of ablation studies on using divergent vs different tokens, as well as using the hidden states, logit, and/or token embeddings as the inputs to the router to show the effectiveness of their choices in the framework. They also report the results on three different datasets and provide comparisons between R2R and the query-level routing methods. Overall, they have a sound experimental setup, in my opinion.

Weaknesses:

(1) I believe the writing can be improved, and it should be revised to correct some sentences. For example, in line 24, I believe "hundreds of parameters" is not correct, as you are saying "LLMs with hundreds of parameters (line 124) and SLMs with "few billions of parameters (line 26)"

(2) Some assumptions, such as "greedy" and "sentence-level" path-following, might be limiting in some cases and for some tasks, but I understand that it may have been necessary to have such assumptions due to the huge search space.

---

> ### Author Rebuttal · Authors · 2025-07-31
>
> We sincerely thank Reviewer gjwQ for the constructive feedback. We appreciate your valuable recognition of the **great significance and thorough experiments** of our approach.
>
> ### [W1, Q2] Typo corrections
>
> > W1: In line 24, I believe "hundreds of parameters" is not correct, as you are saying "LLMs with hundreds of parameters (line 124) and SLMs with "few billions of parameters (line 26)"
>
> We apologize for the typos. To clarify, in line124, we wish to say "LLMs with hundred of **billions of** parameters".
>
> > Q2: Can you please clarify what you mean in lines 170-171? I assumed you would rely on the smaller model for generating the majority of the tokens. Why is it said that only 3% of generated tokens are from the smaller model?
>
> In line 171, the correct statement should read: "...while relying on the **larger Rl-32B** model for only 3% of the generated tokens."
>
> We genuinely thank the reviewer for the careful review. We will carefully proofread the paper to eliminate typos in our writing.
>
> ### [W2.1, L1] Assumptions on greedy decoding
>
> > W2.1: Some assumptions, such as "greedy" and "sentence-level" path-following, might be limiting in some cases and for some tasks, but I understand that it may have been necessary to have such assumptions due to the huge search space.
>
> > L1: Also, the greedy decoding may potentially be problematic for tasks where we want to generate more diverse or creative text.
>
> We thank the reviewer for this insightful question. Below, we provide additional analysis and preliminary experiments on how R2R generalizes to sampling-based decoding strategies.
>
> **Path-Following Routing Under Sampling**
>
> R2R naturally extends to sampling-based decoding by adapting *divergence* labeling to a probabilistic setting.
> Specifically, the deterministic continuation sequence pair $(S_s, S_l)$ from Equations 3–5 are replaced by $k$ **sampled continuation pairs** under a chosen sampling method. *Divergence* is then evaluated for each sampled pair. The routing decision for each token then depends on whether the overall **divergence probability** exceeds a predefined threshold:
> $$
> P(V(S_{s_i}, S_{l_i}) = 0|i \in [0,k)) \geq P_{\text{threshold}}
> $$
>
> If setting $P_{\text{threshold}}$ to the LLM's self-*divergence* probability $P(V(S_l, S_l)=0)$, it ensures the mixed model, under full path-following routing, maintains the same quality expectation as the LLM alone. We will provide detailed descriptions and theoretical analyses of this method in the final manuscript.
>
> **Efficient Data Generation Pipeline**
>
> Sampling multiple continuations for *divergence* labeling is computationally expensive. However, since continuations are sentence-level, we empirically observe that the *divergence* decision primarily depends on the first differing token between the SLM and LLM samples, rather than the subsequent continuations.
> To efficiently approximate probabilistic routing, we therefore only sample for next-token generation $y_i(\theta_s|S_{<i})$ and $y_i(\theta_l|S_{<i})$, while leaving continuations deterministic.
>
> Given limited time for rebuttal and the extensive volume of tokens, we set $k=1$ and $P_{\text{threshold}}=0.5$, simplifying the setup. This approximation incurs overhead comparable to our greedy data generation pipeline.
>
> In implementation, our data-generation pipeline adapts to the new setup by using sampling-based generation for both LLM responses (step 0) and SLM prefill (step 1), while keeping other procedures unchanged.
>
> **Experiment Results**
>
> We adopt DeepSeek-R1’s recommended sampling settings (temperature=0.6, top-p=0.95) for preliminary experiments. We retain the neural router architecture, as the router already accepts embedding of the sampled token.
>
> |Method|Acc. (pass@1)|Param. (B)|Cost (KB)|
> |-|-|-|-|
> |R1-1.5B|27\%|1.5|24.3|
> |R1-32B|61\%|32.0|384.8|
> |-|-|-|-|
> |R1-14B|**58\%**|14.0|170.5|
> |QR-SW|39\%|9.2|135.5|
> |QR-LLM|38\%|9.2|138.6|
> |QR-BERT|40\%|9.4|136.8|
> |QR-MF|41\%|9.1|130.5|
> |R2R|**58\%**|**8.6**|**115.3**|
>
> Under sampling-based decoding, R2R remains Pareto optimal compared to distilled models and query-level routing methods.
>
> ***Divergence* distribution analysis**
>
> |Type|#Query|#Token|#Different|Diff. Rate|#*Divergent*|Div. Rate|
> |-|-|-|-|-|-|-|
> |Math|862|7.3M|858K|11.8\%|438K|6.0\%|
> |Code|987|6.8M|1.2M|17.5\%|627K|9.3\%|
> |QA|805|2.1M|443K|21.3\%|231K|11.1\%|
> |Summary|2654|16.1M|2.5M|15.5%|1.3M|8.0\%|
>
> The average *divergence* rate under sampling slightly increases (+3.1%) but remains low overall, demonstrating consistent *divergence* patterns with greedy decoding.
>
> ### [W2.2, L2] Assumptions on sentence-level path-following routing
>
> > W2.2: Some assumptions, such as "greedy" and "sentence-level" path-following, might be limiting in some cases and for some tasks, but I understand that it may have been necessary to have such assumptions due to the huge search space.
>
> > L2: For example, the SLM might want to generate the sentences in a paragraph in a different order but still follow a valid reasoning path and achieve the same conclusion as the LLM in some cases
>
> We appreciate the reviewer's thoughtful comment. While mislabeling *neutral* continuations as *divergent* typically does not degrade performance, it may occasionally introduce unnecessary computational overhead due to additional LLM activations.
>
> To evaluate mislabeling caused by insufficient continuation, we extend our routing strategy to N-sentence path-following. As N gradually increases from 1 to the total remaining sentences in the LLM-generated response, it transitions smoothly from sentence-level to fully path-following routing.
>
> In experiments on the subset of AIME dataset, we observe that increasing N from 1 to 5 only reduces the divergence rate modestly from 2.91% to 2.66%, corresponding to an average activated parameter reduction of 0.08B. Given this limited improvement, we empirically select N=1 (sentence-level) as our default setting.
>
> ### [Q1] Combining R2R and query-level routing
> > What are your thoughts on combining R2R with some query-level routing methods so that you can choose which SLM or LLM to use based on the difficulty of the query, and then using R2R to route tokens to the SLM or LLM for the generation?
>
> Thank you for the insightful suggestion. We investigated combining R2R with query-level routing methods using the Qwen3 model series with varied sizes, evaluated on the AIME benchmark. Specifically, we treated R2R-S (0.6B+8B) and R2R-L (0.6B+32B) as the SLM and LLM, respectively, for query-level routing.
>
> |Type|Model|Acc.|Param. (B)|Cost (KB)|
> |-|-|-|-|-|
> |SLM|Qwen3-0.6B|12\%|0.6|15|
> |LLM|Qwen3-8B|67\%|8|130|
> |LLM|Qwen3-32B|75\%|32|469|
> |-|-|-|-|-|
> |R2R-S|Qwen3-0.6B + 8B|59\%|2.1|40|
> |R2R-L|Qwen3-0.6B + 32B|67\%|10.2|154|
> |-|-|-|-|-|
> |R2R-M|Qwen3-4B + 8B|65\%|5.4|**95**|
> |-|-|-|-|-|
> |QR-BERT|Qwen3-0.6B + 32B|25\%|5.4|230|
> |QR-LLM|Qwen3-0.6B + 32B|23\%|5.4|238|
> |QR-MF|Qwen3-0.6B + 32B|32\%|5.4|216|
> |QR-SW|Qwen3-0.6B + 32B|30\%|5.4|220|
> |-|-|-|-|-|
> |QR-BERT|R2R-S + R2R-L|63\%|5.4|179|
> |QR-LLM|R2R-S + R2R-L|**67\%**|5.4|175|
> |QR-MF|R2R-S + R2R-L|58\%|5.4|181|
> |QR-SW|R2R-S + R2R-L|62\%|5.4|183|
>
> From these experiments, we conclude:
>
> 1. R2R models generally serve as superior SLM/LLM candidates for query-level routing compared to original single-model setups, consistently advancing the Pareto frontier.
>
> 2. The interaction between medium-sized R2R models and query-level routing involving smaller or larger R2R-LLMs is empirically non-trivial. This observation suggests exciting opportunities for future research into sophisticated routing strategies, which combine the search space of query and token-level routing methods.
>
> We will incorporate these discussions into the revised manuscript, highlighting insights and promising directions for future exploration.

---

> > ### Comment · Reviewer_gjwQ · 2025-08-05
> >
> > Thank you for your detailed response. I appreciate the clear and thorough explanations.

---

> > > ### Author Response · Authors · 2025-08-05
> > >
> > > We sincerely appreciate your thoughtful review and your recognition of our rebuttal and initial submission. We are glad to hear that our explanations effectively addressed your questions and concerns. Thank you once again for your valuable and constructive feedback.

---

### Official Review · Reviewer_uytF · 2025-07-02

**Clarity:** 3
**Significance:** 3
**Originality:** 3
**Rating:** 4
**Confidence:** 4

**Summary:**

The paper proposes R2R (Roads to Rome), a token-level routing method that combines the efficiency of Small Language Models (SLMs) with the reasoning quality of Large Language Models (LLMs). Rather than relying solely on either model, R2R selectively routes divergent tokens to an LLM, while leaving the rest to the SLM. The authors introduce: (1)A token-level data labeling pipeline to identify divergent tokens; (2)A lightweight neural router trained to predict token divergence; (3) A routing strategy that improves inference efficiency with minimal loss in reasoning quality. By applying R2R between DeepSeek R1-1.5B (SLM) and R1-32B (LLM), they show up to 4.6× accuracy improvement over SLM, with 2.8× speedup over the LLM, surpassing distilled and query-level routing methods in performance-cost tradeoffs.

**Questions:**

(1) How does R2R perform under temperature sampling or nucleus sampling? Would the divergence pattern be consistent?
(2) What guarantees does the LLM verifier provide, and could it mislabel semantically divergent tokens as neutral?
(3) Have you considered batch routing decisions (e.g., look-ahead or segment-level) instead of per-token routing?
(4) Could R2R apply to generative tasks like summarization or dialogue, where path divergence may not always be objectively verifiable?

**Ethical Concerns:**

["NO or VERY MINOR ethics concerns only"]

**Limitations:**

Yes.

**Quality:**

3

**Strengths And Weaknesses:**

Strengths:

(1) Novel Insight & Intuition: The paper identifies that most token differences between SLM and LLM are either identical or neutral, and that only a small fraction leads to performance degradation—an important insight for efficient inference.

(2) Efficient Labeling Pipeline: The sentence-level LLM continuation and an LLM verifier to distinguish divergent tokens are well-designed, scalable, and avoid the need for expensive human annotations.

(3) Strong Empirical Results: First, it advances the Pareto frontier on reasoning benchmarks (AIME, GPQA, LiveCodeBench). Second, it achieves impressive speed-accuracy tradeoffs (e.g., 2.8× speedup over R1-32B with comparable accuracy).

(4) Robust Engineering: Implementation details are thorough, including training strategy, threshold tuning, GPU memory optimizations, etc.

Weaknesses:

(1) Verifier Dependence: The data labeling pipeline depends on a separate LLM-based verifier, which may introduce a bias or circular dependency (LLM verifying LLM paths).

(2) Domain Bias in Training Data: The training data (math, QA, and code) may not generalize well to open-ended tasks (e.g., embodied, navigation, etc).

(3) Assumes Greedy Decoding: The method is only validated under greedy decoding. It remains unclear how well R2R generalizes to sampling-based strategies used in many real-world applications (e.g., temperature > 0).

---

> ### Author Rebuttal · Authors · 2025-07-31
>
> We sincerely thank Reviewer uytF for the valuable feedback. We appreciate your great recognition of the **significance, novelty, and strong experimental results** of our approach.
>
> ### [W1, Q2] Verifier depends on LLM, potentially mislabels divergent and neutral tokens.
>
> > W1: The data labeling pipeline depends on a separate LLM-based verifier, which may introduce a bias or circular dependency.
>
> > Q2: What guarantees does the LLM verifier provide, and could it mislabel semantically divergent tokens as neutral?
>
> To evaluate reliability in classifying *divergences* and *neutral* differences, we conducted additional experiments comparing the different LLM verifiers with human experts. We also exame model robustness to mislabeling.
>
> **Verifier Capacity and Verification Quality**
>
> We constructed ground-truth *divergent* labels with **three independent human expert** verifiers [1]. The **general** *divergent* tokens are labeled by majority voting among expert answers, taking up 24.8% of differences. The **core** *divergent* tokens, a subset of highly *divergent* tokens with more strict criteria, are labeled with unanimous consent of experts, which takes up 11.9% of *differences*.
>  he table below compares the chosen verifier (Qwen2.5-72B) against a **fourth independent expert**. Our verifier achieves a comparable performance over human expert, with high recall on core *divergence*.
>
> |Verifier|*Divergent* Rate (%)|Core-Recall (%)|Core-Accuracy (%)|General-Recall (%)|General-Accuracy (%)|
> |-|-|-|-|-|-|
> |Human|35|96|76|84|82|
> |Qwen2.5-72B|35|97|76|88|84|
> |Qwen2.5-7B|39|94|71|85|78|
> |Qwen2.5-3B|34|75|72|69|75|
>
> Note that the verifier judges semantic differences between two continuations, which is notably simpler than generating correct responses or evaluating the entire reasoning path.
> Consequently, even a smaller verifier (Qwen2.5-7B) provides satisfactory performance. However, significantly smaller models (e.g., 3B) may yield less ideal results.
>
> **Verification Quality and Response Quality**
>
> We evaluated how verifier accuracy impacts response quality using sentence-level path-following routing (Equations 3–5) on AIME questions. It evaluates the influence of verifier on final response quality, without incurring the neural router.
>
> As shown below, even a verifier with moderate recall (e.g., 66% for Qwen2.5-3B) can guide better reasoning paths than the baseline R1-7B model.
>
> |Type|Param. (B)|Labeling Verifier|AIME'24 #Acc.@ $\leq$8K|
> |-|-|-|-|
> |LLM|32|-|17|
> |SLM|1.5|-|2|
> |Distill|7|-|8|
> |Path-following routing|2.3|Qwen2.5-72B|16|
> |Path-following routing|2.5|Qwen2.5-7B|12|
> |Path-following routing|2.2|Qwen2.5-3B|11|
>
> Sentence-level path-following routing also connects the verifier prompt to final response quality, enabling direct performance tuning via prompt engineering.
> The current verifier prompt is optimized for Qwen2.5-72B. Empirically, smaller verifiers can benefit from prompts biased towards classifying ambiguous cases as *divergent*, enhancing recall.
> We will include detailed prompt-tuning discussions and extended results in the final manuscript.
>
> **Quality-Guaranteed Verifier**
>
> We also introduce the quality-guaranteed **full path-following routing** (L151-158, proof in Appendix D). For scenarios with shorter reasoning paths and sufficient computational resources, it explicitly ensures the final response quality with generated labels.
>
> [1] We recruited four undergraduate engineering students as annotators. They receive identical contexts and instructions as the verifier LLM, covering 1357 *different* tokens between R1-1.5B to R1-32B on six AIME-2024 questions.
>
>
> ### [W2, Q4] Generalizability across domains and tasks
>
> > W2: The training data (math, QA, and code) may not generalize well to open-ended tasks.
>
> > Q4: Could R2R apply to generative tasks like summarization or dialogue, where path divergence may not always be objectively verifiable?
>
> We appreciate the reviewer’s comments on R2R’s generalizability.
>
> **Generalizable Data Labeling Method**
>
> Our labeling method, relying on semantic comparison rather than formal verification, easily generalizes across tasks. Unlike methods requiring task-specific external tools (e.g., formal proofs [2], code execution [3]), R2R uses an LLM verifier to identify general semantic divergence in *meaning, reasoning, logic, or conclusions* (Appendix E.1).
>
> In practice, we apply the identical labeling strategy and verifier prompt across closed-form math, coding tasks, and open-ended QA tasks. This consistency enables our router to naturally generalize to unseen tasks during inference, as demonstrated later.
>
> For tasks with subjective *divergence* criteria that are challenging to identify semantically within a single sentence, the continuation length can be extended. This adjustment gradually transitions from sentence-level routing to full routing (L151-158), the latter of which only requires overall response quality evaluation—a feasible requirement commonly met for LLM tasks.
>
> **Generalizable Router**
>
> Our router leverages outputs from the SLM (e.g., logits) to predict token *divergence*. Benefiting from the inherent generalizability of SLMs, indicators such as logits entropy robustly identify *divergent* tokens across different tasks.
>
> To validate the router’s generalizability, we directly apply our router, trained for math, QA, and code, to additional benchmarks: Arena-Hard for **Dialog** [4], and MMLU-Redux-**Philosophy**.
>
> |Type|Method|Dialog-Score|Dialog-Param.|Dialog-Cost|Philosophy-Acc.(%)|Philosophy-Param.|Philosophy-Cost|
> |-|-|-|-|-|-|-|-|
> |LLM|R1-32B|5.0|32|65.9|38|1.5|8.7|
> |SLM|R1-1.5B|0.2|1.5|14.3|79|32|32.2|
> |Distill|R1-7B|0.3|7|35.0|57|7|20.6|
> |Distill|R1-14B|2.2|14|56.5|77|14|18.5|
> |R2R|Ours|2.8|6.1|40.9|81|6.7|8.3|
>
> R2R continues to outperform the 14B model with less than 7B parameter size.
>
> [2] X, Huajian, et al. "Deepseek-prover-v1.5: Harnessing proof assistant feedback for reinforcement learning and monte-carlo tree search." arXiv 2024.
>
> [3] D. Jung, et al. “Code execution as grounded supervision for LLM reasoning.” arXiv 2025
>
> [4] We evaluate the first 100 questions in the benchmark to expedite testing.
>
> ### [W3, Q1] Sampling method extension
>
> > W3: The method is only validated under greedy decoding. It remains unclear how well R2R generalizes to sampling-based strategies
>
> > Q1: How does R2R perform under temperature sampling or nucleus sampling? Would the divergence pattern be consistent?
>
> We provide additional analysis and experiments on sampling-based decoding.
>
> **Path-Following Routing Under Sampling**
>
> R2R naturally extends to sampling-based decoding by adapting *divergence* labeling to a probabilistic setting.
> Specifically, the deterministic continuation sequence pair $(S_s, S_l)$ from Equations 3–5 are replaced by $k$ **sampled continuation pairs** under a chosen sampling method. *Divergence* is then evaluated for each sampled pair. The routing decision for each token then depends on whether the overall **divergence probability** exceeds a predefined threshold:
> $$
> P(V(S_{s_i}, S_{l_i}) = 0|i \in [0,k)) \geq P_{\text{threshold}}
> $$
>
> If setting $P_{\text{threshold}}$ to the LLM's self-*divergence* probability $P(V(S_l, S_l)=0)$, it ensures the mixed model, under full path-following routing, maintains the same quality expectation as the LLM alone. We will provide detailed descriptions and theoretical analyses of this method in the final manuscript.
>
> **Efficient Data Generation**
>
> Sampling multiple continuations for *divergence* labeling is computationally expensive. However, since continuations are sentence-level, we empirically observe that the *divergence* decision primarily depends on the first differing token between the SLM and LLM samples, rather than the subsequent continuations.
> To efficiently approximate probabilistic routing, we therefore only sample for next-token generation $y_i(\theta_s|S_{<i})$ and $y_i(\theta_l|S_{<i})$, while leaving continuations deterministic.
>
> Given limited time for rebuttal and the extensive volume of tokens, we set $k=1$ and $P_{\text{threshold}}=0.5$, simplifying the setup. It incurs overhead comparable to our greedy data generation pipeline.
>
> In implementation, our data-generation pipeline adapts to the new setup by using sampling-based generation for both LLM responses (step 0) and SLM prefill (step 1), while keeping other procedures unchanged.
>
> **Experiment Results**
>
> We adopt DeepSeek-R1’s recommended sampling settings (temperature=0.6, top-p=0.95) for preliminary experiments. We retain the neural router architecture, as the router already accepts embedding of the sampled token.
>
> |Method|Acc. (%)(pass@1)|Param. (B)|Cost (KB)|
> |-|-|-|-|
> |R1-1.5B|27|1.5|24.3|
> |R1-32B|61|32.0|384.8|
> |-|-|-|-|
> |R1-14B|**58**|14.0|170.5|
> |QR-SW|39|9.2|135.5|
> |QR-LLM|38|9.2|138.6|
> |QR-BERT|40|9.4|136.8|
> |QR-MF|41|9.1|130.5|
> |R2R|**58**|**8.6**|**115.3**|
>
> Under sampling-based decoding, R2R remains Pareto optimal compared to distilled models and query-level routing methods.
>
> ***Divergence* Distribution Analysis**
>
> |Type|#Query|#Token|#Different|Diff. Rate|#*Divergent*|Div. Rate|
> |-|-|-|-|-|-|-|
> |Math|862|7.3M|858K|11.8\%|438K|6.0\%|
> |Code|987|6.8M|1.2M|17.5\%|627K|9.3\%|
> |QA|805|2.1M|443K|21.3\%|231K|11.1\%|
> |Summary|2654|16.1M|2.5M|15.5%|1.3M|8.0\%|
>
> The average *divergence* rate under sampling slightly increases (+3.1%) but remains low overall, demonstrating consistent *divergence* patterns with greedy decoding.
>
>
> ### [Q3]: Batch routing decisions
>
> > Have you considered batch routing decisions (e.g., look-ahead or segment-level) instead of per-token routing?
>
> Although R2R currently routes at the token-level, it can extend to coarser granularities such as segments: we simply aggregate token‑level signals within a segment and issue a single routing decision. This opens up new trade-offs between routing granularity, promptness, and accuracy. We consider this a promising direction for future work.

---

### Official Review · Reviewer_PS9S · 2025-07-04

**Clarity:** 3
**Significance:** 3
**Originality:** 3
**Rating:** 5
**Confidence:** 4

**Summary:**

This work proposes a routing scheme that delegates the decoding flow from a small LLM (SLM) to a more powerful LLM (LLM) when there is divergence between the reasoning path. A divergence is defined to emerge when there is a disagreement between the predictions of the SLM and LLM and that disagreement will result in a different reasoning path till the end of the sentence. Here the difference between two reasoning paths is evaluated by an external verifier model (Qwen2.5-72B used in this work).

The authors further propose a data generation pipeline to collect labels for such divergent tokens, and then train an simple MLP-based token-level router to dynamically decide when to delegate the reasoning to the LLM. The authors apply the proposed method (R2R) to R1-1.5B and R1-32B models from the DeepSeek family and evaluate on a variety of tasks. The results show impressive performance (comparable to R1-32B and better than R1-14B) with significantly reduced inference costs, which are measured by the number of activated parameters and the wall-clock time (latency and token generation speed).

**Questions:**

- I wonder if MoE can be included in the context here. In my opinion, MoE is another way of utilizing a model with larger capacity. But the difference is that MoE implicitly uses the whole capacity with fixed cost at each token. In contrast, R2R uses small capacity model with low costs most of time but escalate to larger model when needed. It would be great if we can find a way to properly compare the two. I understand that this can be too demanding for the review period. So this is not a request but an open question out of curiosity.

**Ethical Concerns:**

["NO or VERY MINOR ethics concerns only"]

**Final Justification:**

The authors responses addressed my questions and concerns. From my point of view, the authors have evaluated the proposed method with latency and token throughput (# token / sec), where the results are in favor. I look forward to more discussion and hopefully numerical comparison with more related works in the final version.

**Limitations:**

yes

**Quality:**

3

**Strengths And Weaknesses:**

Strengths:
+ The intuition of the dynamic reasoning makes sense and the definition of divergent tokens is reasonable.
+ The empirical results are impressive and the decoding acceleration is practically promising.
+ The authors provide ablation study on the design choice of the MLP router in Section 5.4. While the results are insufficiently discussed (see comments below), the study does support the choice of the input fields to the router and the training objective.
+ The paper is written clearly with plenty of details in support.

Weaknesses:
Just to clarify that I think this work is of good quality overall. So many of the weaknesses are not hard criticisms but constructive feedback on which I would invite more discussions with the authors.
- (Discussion against [3]) I understand that [3] is concurrent. But now that the authors mention [3] in Line 227, I think it would be better to discuss properly in the section of related work.
- (Insufficient discussion of ablation study) I think the ablation study on the routing objective is not sufficiently discussed. I suppose delegate "more" tokens to LLM will not hurt the performance this much. Is it because the objective change causes the router misses more divergent tokens? Moreover, it is not clear what the authors mean by "it fails to reach the original accuracy within the same amount of LLM usage, ..." Does it imply that the authors manually control the activation of the LLM in this experiment?

---

> ### Author Rebuttal · Authors · 2025-07-31
>
> We sincerely thank Reviewer PS9S for the extensive feedback. We appreciate your valuable recognition of the **intuition, quality and impressive empirical results** of our approach.
>
> ### [W1] Discussion on concurrent work (SplitReason)
>
> > I understand that [3] is concurrent. But now that the authors mention [3] in Line 227, I think it would be better to discuss properly in the section on related work.
>
> Thank you for the great suggestion. We will include a proper discussion of the concurrent work SplitReason in our revised manuscript. As the SplitReason repository is under maintenance during the rebuttal period, we plan to include the experimental comparison in the final version.
>
> Both R2R and SplitReason explore mixed inference strategies combining small and large language models, but differ in routing granularity and objectives.
>
> SplitReason aims to offload **difficult reasoning steps** to the LLM. Specifically, it first uses a strong LLM to identify challenging reasoning steps, then trains the SLM to generate a special token (i.e., `<bigmodel>`) signaling the LLM to take over these difficult steps.
>
> In contrast, R2R addresses **token-level *divergences*** rather than step-level difficulty. Different from SplitReason, R2R also targets the subtle scenario where the SLM and LLM may agree on challenging steps, yet diverge unexpectedly on seemingly straightforward tokens. Such *divergences* can significantly alter the subsequent reasoning path, thus requiring immediate correction.
>
> We believe the two complementary perspectives would offer comprehensive contexts for efficient reasoning, inspiring even more effective mixed inference future works.
>
>
>
> ### [W2] Insufficient discussion of ablation study
>
> > I think the ablation study on the routing objective is not sufficiently discussed.
>
> Thank you for the thoughtful questions. We clarify as follows:
>
> > I suppose delegate "more" tokens to LLM will not hurt the performance this much.
>
> 1. **More LLM tokens do not commonly hurt AIME performance.**
>
>     Indeed, all 7 AIME questions correctly answered by the SLM are also correctly answered by the LLM, confirming that the LLM consistently outperforms the SLM rather than specializing in different types of math questions.
>
> > Is it because the objective change causes the router misses more divergent tokens?
>
> 2. **Yes, the *different* objective leads to more missed *divergent* tokens.**
>
>     To investigate this, we evaluate the *divergent* token recall of different routers. We use ground-truth responses and *divergence* labels generated with our data pipeline on AIME. Routers trained with *different* and *divergent* objectives are marked at Router1 and Router2, respectively.
>
> |LLM Rate|Router1-Recall|Router2-Recall|
> |-|-|-|
> |5%|0.50|0.28|
> |10%|0.75|0.46|
> |11%|0.77|0.51|
> |12%|0.80|0.55|
> |13%|0.83|0.59|
> |14%|0.85|0.63|
> |15%|0.87|0.66|
> |16%|0.90|0.69|
> |20%|0.94|0.79|
>
> As demonstrated in the table, the router trained with the *different* objective shows lower recall, even at higher LLM usage. This is because it wastes LLM calls on *neutral* differences rather than targeting truly *divergent* tokens.
>
> > Moreover, it is not clear what the authors mean by "it fails to reach the original accuracy within the same amount of LLM usage, ..." Does it imply that the authors manually control the activation of the LLM in this experiment?
>
> 3. **Yes, we manually adjust the routing threshold to control LLM activation.**
>
>     Since R2R allows for flexible trade-offs between LLM usage and performance, each router defines a trade-off curve. For ablation comparison, we adjust thresholds to match or slightly exceed the LLM rate of our main router, and then compare their accuracies. Despite higher LLM usage, the routers with  objective or reduced inputs fail to reach comparable accuracy.
>
> We will revise the manuscript to improve the clarity of this ablation discussion.
>
>
>
> ### [Q1] Including MoE in the context
>
> > I wonder if MoE can be included in the context here. In my opinion, MoE is another way of utilizing a model with larger capacity. But the difference is that MoE implicitly uses the whole capacity with fixed cost at each token. In contrast, R2R uses small capacity model with low costs most of time but escalate to larger model when needed. It would be great if we can find a way to properly compare the two. I understand that this can be too demanding for the review period. So this is not a request but an open question out of curiosity.
>
> Thank you for raising this interesting perspective. Below, we first present a conceptual comparison between MoE and R2R, followed by empirical comparisons, and finally discuss how the two approaches can complement each other to further improve the efficiency-accuracy trade-off.
>
> **Conceptual Comparison**
>
> |Design Aspect|MoE|R2R|
> |-|-|-|
> |Partial activation|Yes|Yes|
> |Routing granularity|Fine-grained (Experts)|Coarse-grained (Models)|
> |Subject model sizes|Equal parameters per expert|Different parameters per model|
> |Training overhead|Full training from scratch|Router training only (SLM+LLM pre-existing)|
> |Supervision|Next-token predictions|Explicit routing decisions|
>
> As noted by the reviewer, both MoE and R2R leverage partial activation of larger capacity models, but with distinct granularities, subject sizes, training overhead, supervision and complementary insights. This conceptual distinction motivates their integration, which we explore later.
>
> **Empirical Comparison**
>
> We evaluate the MoE model Qwen3-30BA3B and the R2R model (combining Qwen3-0.6B and Qwen3-8B) on the AIME benchmark. The 60 questions are split into Easy (28) and Hard (32) subsets based on whether the question ID $\leq$ 7.
>
> |Type|Models|Easy-Accuracy|Easy-Parameters (B)|Hard-Accuracy|Hard-Parameters (B)|
> |-|-|-|-|-|-|
> |R2R|Qwen3-0.6B + Qwen3-8B|93\%|2.9|41\%|2.6|
> |MoE|Qwen3-30BA3B|93\%|3.3|59\%|3.3|
>
> These results highlight the strengths of each approach. Thanks to extensive and sophisticated pretraining, state-of-the-art MoE models can match dense model performance (e.g., both Qwen3-30BA3B and Qwen3-32B achieve 75% accuracy on AIME in our evaluation). However, their fixed per-token overhead limits efficiency for simpler tasks.
>
> In contrast, R2R dynamically adjusts computation based on token-level *divergence*. It achieves lower cost on easier examples by relying more on the SLM, but its small router and much lighter training cannot match MoE performance on harder questions with the same activated parameters.
>
> Given these complementary advantages, we explore the combination of both methods.
>
> **Combining MoE and R2R**
>
> *MoE for R2R*
>
> R2R is directly compatible with efficiency-optimized models such as MoE models. We validate R2R performance using the Qwen3-30BA3B MoE as the LLM:
>
> | |Acc.|Param. (B)|Cost (KB)|
> |-|-|-|-|
> |Qwen3-0.6B|12\%|0.6|14.7|
> |Qwen3-1.7B|37\%|1.7|32.8|
> |-|-|-|-|
> |Qwen3-32B|75\%|32.0|469.1|
> |R2R (0.6B+32B)|67\%|10.2|154.0|
> |-|-|-|-|
> |Qwen3-30BA3B (MoE)|75\%|3.0|50.5|
> |R2R (0.6B+30BA3B MoE)|68\%|1.3|21.7|
>
> Combining R2R with MoE models significantly improves the Pareto frontier, achieving robust accuracy even at extremely low activated parameters per token.
>
> *R2R for MoE*
>
> R2R insights can also enhance MoE model designs.
> Current MoE methods generally use uniform-sized experts. Introducing mixed-sized experts like R2R could enable query-adaptive expert activation.
> Moreover, supplementing MoE's token prediction objective with explicit *divergence* or difficulty-based routing supervision from R2R may encourage adaptive usage of larger experts only for critical decoding steps.
> Although this approach would require retraining MoE models and is beyond the scope of this paper, it opens exciting future research directions.
>
> We will incorporate these discussions into the revised manuscript, providing insights for further exploration.

---

> > ### Author Response · Authors · 2025-08-06
> >
> > Dear Reviewer PS9S,
> >
> > We have provided responses to all your concerns in our rebuttal. We would greatly appreciate it if you could kindly let us know whether our replies have adequately addressed your concerns. If there are any points that remain unclear or could benefit from further clarification, we would be glad to discuss them further.
> >
> > Thank you once again for your valuable feedback and for taking the time to review our submission.
> >
> > *(We are following up in line with the recent PC guidance that welcomes authors to initiate discussions.)*

---

### Official Review · Reviewer_ZxyV · 2025-07-04

**Clarity:** 3
**Significance:** 3
**Originality:** 2
**Rating:** 5
**Confidence:** 4

**Summary:**

This paper proposes Roads to Rome (R2R), a token-level routing method that selectively utilizes LLMs only for tokens that cause reasoning path divergence when generated by small language models (SLMs). The core insight is that only ~11% of tokens genuinely diverge reasoning paths between SLMs and LLMs, with most differences being either identical or neutral variations (e.g., abbreviations, alternative expressions). The authors develop an automatic data generation pipeline to identify divergent tokens and train a lightweight router for real-time routing decisions. Experiments on DeepSeek R1 models demonstrate 2.8X speedup over R1-32B while maintaining comparable performance, and superior efficiency compared to query-level routing and distilled models.

**Questions:**

1. **Sampling method extension**: How fundamental is the greedy decoding limitation? Could the neutral/divergent distinction be meaningfully applied to temperature-based sampling? This seems critical for practical deployment - can you provide any analysis or preliminary experiments?
2. **Cross-model generalization**: Would a router trained on DeepSeek R1-1.5B/32B work with other model pairs? How much retraining would be required?
3. **Verifier robustness analysis**: How sensitive is performance to verifier quality? What happens with weaker verifier models? Could you provide human evaluation of neutral/divergent distinctions to validate the verifier's reliability?

4. **Systematic failure modes**: Can you identify task types or reasoning patterns where the approach consistently fails? Does performance degrade on highly creative or non-linear reasoning tasks?

5. **Real-world deployment considerations**: How would this integrate with existing serving infrastructure? What are the latency implications of token-level routing decisions in production systems?

**Ethical Concerns:**

["NO or VERY MINOR ethics concerns only"]

**Final Justification:**

The authors did many new experiments and analyses, and addressed all my concerns (and also addressed most of the concerns of the other reviewers)

**Limitations:**

The authors adequately acknowledge the greedy sampling restriction and need for system optimizations. However, they significantly understate the generalizability concerns and don't sufficiently address the verifier dependency issues. The authors should also discuss the computational overhead more transparently in the context of alternative approaches.

**Quality:**

3

**Strengths And Weaknesses:**

**Strengths:**
1. **Novel and well-motivated insight**: The distinction between neutral and divergent token differences is conceptually sound and empirically well-supported. The observation that most SLM-LLM differences don't affect reasoning paths is valuable.
2. **Comprehensive experimental evaluation**: The paper evaluates across several benchmarks (AIME, GPQA, LiveCodeBench) with appropriate baselines including query-level routing, distilled models, and speculative decoding. They also measured real hardware speedup.
 3. **Strong empirical results**: Achieving 92% of R1-32B accuracy with only 17% average parameters is impressive. The 4.6× accuracy improvement over R1-1.5B with minimal LLM usage demonstrates practical value.

**Weaknesses:**

1. **Critical limitation to greedy decoding**: Most production systems use temperature-based sampling or other stochastic methods. The authors acknowledge this but underestimate its significance for deployment.
2. **Limited generalizability**: All experiments use only DeepSeek R1 models. No evidence that the approach generalizes across model families, architectures, or even different size pairs within the same family.

3. **Shallow failure analysis**: Limited discussion of systematic failure modes or scenarios where neutral/divergent distinction breaks down. When does the approach perform poorly, and why?

---

> ### Author Rebuttal · Authors · 2025-07-31
>
> We sincerely thank Reviewer ZxyV for the extensive feedback. We appreciate your valuable recognition of the **novelty, value and strong experimental results** of our approach.
>
> ### [W1, Q1]: Assumes greedy decoding, should extend to other sampling methods
>
> > W1: Critical limitation to greedy decoding: Most production systems use temperature-based sampling or other stochastic methods.
>
> > W2: Sampling method extension: How fundamental is the greedy decoding limitation? Could the neutral/divergent distinction be meaningfully applied to temperature-based sampling?
>
> We summarize the formulation, data generation pipeline, and experimental results of R2R under sampling-based decoding.
> Due to word limit, please refer to our response to reviewer gjwQ, [W2.1,L1] for additional details.
>
> **Divergent Label and Data Generation Under Sampling**
>
> R2R naturally extends to sampling-based divergence by extending the verifier function (Equation 3) under probabilistic setup. The current token is *divergent* if the divergence probability (under deterministic verification) for SLM and LLM continuation samples exceeds the LLM’s intrinsic self-divergence.
>
> Observing that divergence primarily depends on the first differing token between SLM and LLM samples (rather than subsequent continuations), our data-generation pipeline easily adapts to sampling. Specifically, it uses sampling-based generation for LLM generation (step 0) and SLM prefill (step 1), while keeping other procedures unchanged.
>
> **Experiment Results**
>
> We adopt DeepSeek-R1’s recommended sampling settings (temperature=0.6, top-p=0.95) for experiments.
>
> |Method|Acc. (pass@1)|Param. (B)|Cost (KB)|
> |-|-|-|-|
> |R1-1.5B|27\%|1.5|24.3|
> |R1-32B|61\%|32.0|384.8|
> |-|-|-|-|
> |R1-14B|**58\%**|14.0|170.5|
> |QR-SW|39\%|9.2|135.5|
> |QR-LLM|38\%|9.2|138.6|
> |QR-BERT|40\%|9.4|136.8|
> |QR-MF|41\%|9.1|130.5|
> |R2R|**58\%**|**8.6**|**115.3**|
>
> Under sampling-based decoding, R2R remains Pareto optimal compared to distilled models and query-level routing methods.
>
> ### [W2]: Generalizability across model families, architectures, and size pairs
>
> > All experiments use only DeepSeek R1 models. No evidence that the approach generalizes across model families, architectures.
>
> We extend our experiments to evaluate generalizability across:
>
> 1. **Model families** (Qwen3 in addition to DeepSeek-R1)
>
> 2. **Architectures** (dense vs. MoE)
>
> 3. **Size pairs** (including 0.6B, 1.7B, 8B, 30B-A-3B, and 32B)
>
> We report accuracy on the AIME benchmark using the same setup as Table 2.
>
> |Type|Model(s)|Acc. |Param. (B)|Cost (KB)|
> |-|-|-|-|-|
> |SLM|Qwen3-0.6B|12\%|0.6|15|
> |SLM|Qwen3-1.7B|37\%|1.7|33|
> |LLM|Qwen3-8B|67\%|8.0|130|
> |LLM|Qwen3-30BA3B (MoE)|75\%|3.0|51|
> |-|-|-|-|-|
> |Distill|Qwen3-4B|62\%|4.0|71|
> |QR-SW|0.6B+8B|47\%|4.1|85|
> |QR-LLM|0.6B+8B|52\%|4.0|81|
> |QR-BERT|0.6B+8B|48\%|4.0|77|
> |QR-MF|0.6B+8B|45\%|4.1|79|
> |QR-SW|1.7B+8B|49\%|4.0|75|
> |QR-LLM|1.7B+8B|50\%|4.0|72|
> |QR-BERT|1.7B+8B|50\%|4.1|73|
> |QR-MF|1.7B+8B|55\%|4.1|73|
> |-|-|-|-|-|
> |R2R|0.6B+8B|65\%|2.7|49|
> |R2R|1.7B+8B|68\%|3.7|62|
> |R2R|0.6B+30BA3B (MoE)|70\%|1.5|25|
>
> These results demonstrate that R2R generalizes well across model families, architectures, and various size pairs.
> Notably, R2R constantly outperforms distilled models and query-level routing baselines, while activating fewer parameters ($\leq$ 3.7B on average).
>
> We will extend Figure 5 in the revised manuscript to include the full cost–accuracy trade-offs across these settings, allowing readers to better visualize how R2R's Pareto front shifts under different model combinations and routing thresholds.
>
> ### [W3, Q4]: Systematic failure analysis
>
> > W3: Limited discussion of systematic failure modes or scenarios where neutral/divergent distinction breaks down.
>
> > Q4: Can you identify task types or reasoning patterns where the approach consistently fails? Does performance degrade on highly creative or non-linear reasoning tasks?
>
> To identify systematic failure modes, we analyzed the response status of R2R on AIME and GPQA benchmarks. As shown below, the primary failure mode occurs when R2R cannot complete reasoning within the 32K token limit. This typically happens due to repetitive reasoning patterns not encountered during router training.
> Additionally, R2R tends to invoke the LLM slightly more frequently on the GPQA benchmark, particularly for queries involving rare tokens such as complex protein or chemical compound names.
>
> |BenchMark|Correct|Unfinished|Wrong-same answer with LLM/SLM|Wrong-different answer with LLM/SLM|
> |-|-|-|-|-|
> |AIME (60)|33|25|0/0|2/2|
> |GPQA (198)|87|85|6/3|20/23|
> |Total (258)|120|110|6/3|22/25|
>
> We will include a more detailed analysis of these failure cases in the final manuscript.
>
> ### [Q2] Cross-model generalization of the router
>
> > Would a router trained on DeepSeek R1-1.5B/32B work with other model pairs? How much retraining would be required?
>
> We wish to clarify that the R2R router is designed to be trained per SLM-LLM pair, requiring about 448 GPU hours for initial data generation and 2 GPU hours for router training.
> This overhead remains significantly lower than the distillation processes commonly employed by SOTA models. The cross-model generalizability under this setup is evaluated in [W2].
>
> | |Models|Router Training Data|AIME Accuracy|Param. (B)|Cost (KB)|
> |-|-|-|-|-|-|
> |SLM|0.6B|-|11.7\%|0.6|15|
> |LLM1|8B|-|66.7\%|8|130|
> |LLM2|32B|-|75.0\%|32|469|
> |R2R|0.6B+8B|0.6B+8B|65.0\%|2.69|49|
> |R2R|0.6B+8B|0.6B+32B|53.3\%|2.61|50|
>
> Nevertheless, our empirical results indicate acceptable performance when directly substituting the LLM of a trained router without additional retraining. This can be attributed to the common *divergence* patterns shared by strong LLMs paired with the same weaker SLM.
>
> ### [Q3] Verifier robustness analysis
>
> > How sensitive is performance to verifier quality? What happens with weaker verifier models? Could you provide human evaluation of neutral/divergent distinctions to validate the verifier's reliability?
>
> **Human Evaluation**
>
> We conducted a human evaluation to validate the verifier's reliability. Specifically, we recruited four undergraduate students to independently label 1,357 *differences* as either *neutral* or *divergent*. The *differences* are between R1-1.5B and 32B, on the first six AIME-2024 questions. Three annotators' labels determined ground-truth *divergence* by unanimous consent (core *divergence*) and majority voting (general *divergence*). The fourth annotator's labels were compared with ground truth. Annotators and LLM verifiers received identical contexts and instructions. Our selected verifier (Qwen2.5-72B) closely matches human expert performance:
>
> |Verifier|*Divergent* Rate (%)|Core-Recall (%)|Core-Accuracy (%)|General-Recall (%)|General-Accuracy (%)|
> |-|-|-|-|-|-|
> |Human|35|96|76|84|82|
> |Qwen72B|35|97|76|88|84|
> |Qwen7B|39|94|71|85|78|
> |Qwen3B|34|75|72|69|75|
>
> **Verification Quality**
>
> To examine sensitivity to verifier accuracy, we applied our sentence-level path-following routing method (Equations 3–5) across verifiers. Our analysis reveals that even moderate-recall verifiers (e.g., Qwen2.5-3B with 66% recall) still significantly improve reasoning path guidance compared to the baseline (R1-7B).
>
> |Type|Model Param. (B)|Labeling Verifier|AIME'24 #Acc.@ $\leq$8K|
> |-|-|-|-|
> |LLM|32|-|17|
> |SLM|1.5|-|2|
> |Distill|7|-|8|
> |Path-following routing|2.3|Qwen2.5-72B|16|
> |Path-following routing|2.5|Qwen2.5-7B|12|
> |Path-following routing|2.2|Qwen2.5-3B|11|
>
> Note that the current verifier prompt is tuned for Qwen2.5-72B verifier. Empirically, smaller verifiers should benefit from prompts biased towards classifying ambiguous cases as *divergent*, enhancing recall. We will include detailed prompt-tuning procedures and extended results in the final manuscript.
>
> ### [Q5] Real-world deployment considerations
>
> > How would this integrate with existing serving infrastructure? What are the latency implications of token-level routing decisions?
>
> We integrated R2R into the production-level SGLang serving infrastructure, as evidenced by the high absolute throughput results of our method and baselines.
>
> To assess latency implications of R2R, we performed a detailed runtime breakdown of R2R using two A800 GPUs on the AIME benchmark, averaged over 60 questions:
>
> |Component|Percentage of Total Time|
> |-|-|
> |Router|5.96%|
> |LLM(R1-32B)|64.97%|
> |SLM(R1-1.5B)|26.77%|
> |Others|2.30%|
>
> This indicates that token-level routing decisions introduce minimal latency overhead. Additionally, given the small size of the neural router (56MB), we anticipate further runtime improvements through additional system-level optimizations.
>
> ### [L1] Discuss computational overhead
>
> > The authors should also discuss the computational overhead more transparently in the context of alternative approaches.
>
> Following prior work [1, 2], we analyze both FLOPs and memory access overhead on the AIME benchmark across different approaches.
>
> |Model|Total Computation (TFLOPs)|Total Memory Access (TB)|Avg. Memory Access Per Token (GB)|
> |-|-|-|-|
> |R1-1.5B|157|104|3.7|
> |R1-14B|630|503|29.4|
> |R1-32B|1136|966|63.6|
> |Eagle2|25490|515|29.6|
> |HASS|26809|544|28.9|
> |R2R|1502|216|11.7|
>
> Given the memory-bound nature of decoding, memory access significantly impacts throughput. R2R notably reduces memory access compared to R1-32B with only a modest increase in computation, primarily due to additional KV-cache updates from switching between the LLM and SLM.
> In contrast, speculative decoding approaches (e.g., Eagle2, HASS) incur substantially higher computational costs from their tree-structured draft and verification processes (60 tokens per cycle). Despite this overhead, they achieve significant throughput improvements through reduced memory access during decoding.
>
> [1] Hoffmann, Jordan, et al. "Training compute-optimal large language models." NeurIPS. 2022.
>
> [2] Yang, Fan, et al. "GLITCHES: GPU-FPGA LLM Inference Through a Collaborative Heterogeneous System." HPEC. 2024.

---

> > ### Author Response · Authors · 2025-08-06
> >
> > Dear Reviewer ZxyV,
> >
> > We have provided responses to all your concerns in our rebuttal. We would greatly appreciate it if you could kindly let us know whether our replies have adequately addressed your concerns. If there are any points that remain unclear or could benefit from further clarification, we would be very glad to discuss them further.
> >
> > Thank you once again for your valuable feedback and for taking the time to review our submission.
> >
> > *(We are following up in line with the recent PC guidance that welcomes authors to initiate discussions.)*

---

> > ### Comment · Reviewer_ZxyV · 2025-08-07
> >
> > HI, thanks for the detailed response.
> > You addressed my concerns.

---

> > > ### Author Response · Authors · 2025-08-07
> > > **Appreciation for Your Follow-Up**
> > >
> > > We sincerely appreciate your thoughtful review and continued recognition of our submission and rebuttal. We're glad to hear that our responses have fully addressed your concerns. Thank you once again for your valuable and constructive feedback.

---

### Note · Authors · 2025-08-13

We thank reviewers and AC for the valuable feedback and time. We appreciate consistent recognition across initial reviews and rebuttal, highlighting R2R’s **novel insight** (ZxyV, uytF), **practical significance** (all reviewers), and **strong empirical results** (all reviewers). Below we reiterate the key points:

---

**Key recognitions:**

**1. Novel insight.**

R2R distinguishes neutral and divergent differences, revealing that only ~5% of tokens genuinely diverge reasoning paths between the SLM and LLM.
Reviewers found this insight **novel, well-motivated, and valuable for efficient inference.**

**2. Strong significance.**

R2R enables **more efficient performance scaling at test-time** by routing only divergent tokens to the LLM, advancing the performance–efficiency **Pareto frontier** across AIME, LiveCodeBench, and GPQA.

**3. Impressive experimental results.**

At 5.6B avg. parameters, R2R beats R1-7B accuracy by **1.6x**, even exceeding R1-14B by 5%, achieving **1.6-2.8x wall-clock speedups** over R1-14B/32B.

**4. Broader impact.**

R2R reveals wide **prediction consistency** and pinpoints **truly divergent tokens** between SLM and LLM, offering a lens for **cost-adaptive** reasoning systems.

---

**Key rebuttal additions:**

**1. Extend sampling method.**

Beyond greedy decoding, we extend R2R to **nucleus (top-p) with temperature**. R2R still greatly advances Pareto frontier, with **1.4-1.5x** higher AIME score over query-level routing, **reaching R1-14B** score with only **8.6B** avg. parameters.

**2. Extend model family, architecture, and sizes.**

Beyond R1, we evaluate on **Qwen3** (dense & MoE) across 0.6B, 1.7B, 8B, 30B-A3B, and 32B. R2R consistently outperforms base Qwen3 and query-level routing, showing strong generalizability.

**3. Extend evaluation tasks.**

Beyond math, QA, and coding, we add Arena-Hard (**dialog**), and MMLU-Redux-**Philosophy**. R2R continues to **outperform R1-14B** with only **6.1-6.7B** avg. parameters.

**4. Verifier evaluation with human annotation.**

With four annotators labeling 1357 tokens, we validate the **human-expert-level** reliability of R2R verifier. We also confirm its robustness with end-to-end tests across **3–72B** verifiers.

**Additional analyses.**

We also add results on **router cross-model generalization** (ZxyV), **FLOPs and memory access** (ZxyV), **combining R2R and MoE** (PS9S), and **combining R2R with query-level routing** (gjwQ).

We will add these in the final version.

---

### Decision · Program_Chairs · 2025-09-17

**Decision:**

Accept (poster)

**Comment:**

This paper generates tokens rapidly from a small language model (SLM), and uses the state of the SLM to predict at each token whether a large language model (LLM) might have done something different, in which case it consults the LLM.  This is somewhat similar to speculative decoding, except that in speculative decoding the LLM checks every predicted token, whereas here it checks only the ones that are predicted to be divergent.  This may introduce errors but it also allows tolerance of unimportant divergences between the two models.

**Strengths:** A plausible and interesting method with careful evaluation and demonstrated improvements to the Pareto frontier.

**Weaknesses:** The reviewers (and the AC) identified some minor issues, but the only serious weakness raised was the restriction to greedy decoding, which the authors relaxed in the rebuttal.

The other main weakness is that this is not simply a drop-in decoding method.  The predictor that decides whether it is worth calling the LLM requires training data that is expensively generated with the help of a domain-dependent verifier.  Such a predictor may not generalize to new domains.  (Even in-domain, the training method for the predictor [seems suboptimal](https://openreview.net/forum?id=DpeJYRFRQY&noteId=x5Nqh3v9So).)

**Rationale:** Good consensus among reviewers.

**Rebuttal period:** The authors were very responsive and provided many new experiments, which were appreciated by the reviewers.

-------
# Questions from AC during AC-reviewer discussion period that may be useful to authors

## connections to related work

Are they evaluating cost appropriately?  If I understand correctly, they generate an SLM token at *every* timestep.  Then they probe the SLM's state to decide whether an LLM would have been better.  If so, they append all previously generated SLM tokens to the LLM's KV cache so that the LLM can now make a prediction, and they use that prediction instead.

This means that a sequence like SSSSSLLLLLSS will be efficient.  (The initial SSSSS can be encoded in parallel when they are appended to the LLM context to generate the first L; the LLLLL are appended automatically as they are decoded; and the final SS don't need to be appended at all.)

But a sequence like SLSLSLSLSLSS has many switches to L, and hence less parallelism.  Until the final SS, each S must be separately appended to the LLM context in order to generate the subsequent L.

To distinguish between these sequences, shouldn't they evaluate something like latency, not just activated parameter count (or FLOPS)?   And shouldn't their classifier be optimized to minimize that metric, meaning that it should prefer to reduce the number of switches to L (other things equal)?

## cost evaluation

Are they evaluating cost appropriately?  If I understand correctly, they generate an SLM token at *every* timestep.  Then they probe the SLM's state to decide whether an LLM would have been better.  If so, they append all previously generated SLM tokens to the LLM's KV cache so that the LLM can now make a prediction, and they use that prediction instead.

This means that a sequence like SSSSSLLLLLSS will be efficient.  (The initial SSSSS can be encoded in parallel when they are appended to the LLM context to generate the first L; the LLLLL are appended automatically as they are decoded; and the final SS don't need to be appended at all.)

But a sequence like SLSLSLSLSLSS has many switches to L, and hence less parallelism.  Until the final SS, each S must be separately appended to the LLM context in order to generate the subsequent L.

To distinguish between these sequences, shouldn't they evaluate something like latency, not just activated parameter count (or FLOPS)?   And shouldn't their classifier be optimized to minimize that metric, meaning that it should prefer to reduce the number of switches to L (other things equal)?